# In vivo three-dimensional multispectral photoacoustic imaging of dual enzyme-driven cyclic cascade reaction for tumor catalytic therapy

Shan Lei [1,2,3], Jing Zhang[1,3], Nicholas Thomas Blum[1], Meng Li[1], Dong-Yang Zhang[1], Weimin Yin[1], Feng Zhao[1], Jing Lin[1] & Peng Huang [1✉]

Non-invasive visualization of dynamic molecular events in real-time via molecular imaging may enable the monitoring of cascade catalytic reactions in living systems, however effective imaging modalities and a robust catalytic reaction system are lacking. Here we utilize three-dimensional (3D) multispectral photoacoustic (PA) molecular imaging to monitor in vivo cascade catalytic therapy based on a dual enzyme-driven cyclic reaction platform. The system consists of a two-dimensional (2D) Pd-based nanozyme conjugated with glucose oxidase (GOx). The combination of nanozyme and GOx can induce the PA signal variation of endogenous molecules. Combined with the PA response of the nanozyme, we can simultaneously map the 3D PA signals of dynamic endogenous and exogenous molecules associated with the catalytic process, thus providing a real-time non-invasive visualization. We can also treat tumors under the navigation of the PA imaging. Therefore, our study demonstrates the imaging-guided potential of 3D multispectral PA imaging in feedback-looped cascade catalytic therapy.

[1] Marshall Laboratory of Biomedical Engineering, International Cancer Center, Laboratory of Evolutionary Theranostics (LET), School of Biomedical Engineering, Shenzhen University Health Science Center, Shenzhen 518060, China. [2] Key Laboratory of Optoelectronic Devices and Systems of Ministry of Education and Guangdong Province, College of Physics and Optoelectronic Engineering, Shenzhen University, Shenzhen 518060, China. [3] These authors contributed equally: Shan Lei, Jing Zhang. ✉email: peng.huang@szu.edu.cn

In vivo catalytic processes, such as bioorthogonal catalysis, enzyme or nanozyme-mediated catalytic reactions and enzyme-instructed self-assembly behaviors are of great promise for biomedical applications[1–5]. Significantly, these catalytic processes are often accompanied by the occurrence of multi-molecular events, thus non-invasive visualization of these events can deepen our understanding of biological processes. Among current catalytic processes, cascade catalytic reactions mediated by nanozymes can efficiently catalyze the production of cytotoxic reactive oxygen species (ROS)[6–9] that disrupt the intracellular adaptation mechanisms via oxidative stress[10], holding tremendous potential in catalytic therapies[11]. During the catalytic process, the biological system and nanozymes often involve the dynamic variation of molecular concentrations, including the variations of endogenous molecules (such as oxyhemoglobin (OxyHb) and deoxyhemoglobin (DeoxyHb)) and exogenous molecules (such as therapeutic or contrast agents). The synchronously non-invasive measurement of local dynamic information of molecular events can reflect the state of physiology and pathology at the molecular level[12,13], thus advancing the feasibility of nanozymes for in vivo clinical diagnosis or therapy. Although, current molecular imaging modalities, such as optical imaging[14–17], single-photon emission computed tomography (SPECT)[18], magnetic resonance imaging (MRI)[19,20], ultrasound (US)[21,22], computed tomography (CT)[13] and positron emission tomography (PET)[23,24], have afforded individual or multimodal imaging of molecular events. Most of these imaging modalities are hard to synchronously monitor the feedback of multiple dynamic molecular events in vivo over time. Therefore, real-time non-invasive imaging techniques are urgent to understand the cascade catalytic therapy.

Photoacoustic (PA) imaging, a non-ionizing hybrid imaging modality[25], which is capable of providing the strong optical contrast and high ultrasonic spatial resolution in deep scattering media, overcoming the drawbacks of pure optical and US imaging[26]. Furthermore, the greatest advantage of PA imaging over other imaging modalities is the capability of label-free imaging on all scales[27–30]. For example, label-free PA imaging can evaluate simultaneously both OxyHb and DeoxyHb in vivo[31]. Of note, the PA signals of these endogenous molecules show little variation in many physiological and pathological processes[32]. Therefore, during the pathological processes of cascade catalytic therapy, if the exogenous nanozymes break the current states of endogenous molecules, which can undergo an intrinsic signal evolution, thus providing a correlation between nanozyme states and the occurrence of molecular events in real-time. To achieve this goal, we need to develop a simple and robust nanozyme-based cascade catalytic system. In this regard, the excellent PA property and effective catalytic activity are the main prerequisite for nanozymes. Additionally, in order to maximize the imaging accuracy and spatial resolution, the three-dimensional (3D) PA imaging highlights the potential to map the PA signals in the targeted tissues[33,34]. However, 3D PA imaging has not yet been employed for monitoring the multi-molecular events during the cascade catalytic therapy.

Towards this goal, two enzymes are used to construct a biodegradable cyclic cascade catalytic system (denoted as PMNSG). One is glucose oxidase (GOx), which can produce hydrogen peroxide ($H_2O_2$) through the oxidation of glucose[35] to promote nanozyme-mediated catalytic reaction. The other is a two-dimensional (2D) PdMo bimetallene nanosheet (PMNS)[36], a promising nanozyme, which not only exhibits high catalase (CAT)-like activity under neutral or acidic conditions[37], promoting the catalytic decomposition of $H_2O_2$ into $O_2$, thus potentiating the catalytic process of GOx, reducing the systemic toxicity of GOx and inducing the PA signals variation of OxyHb and DeoxyHb, but also shows peroxidase (POD)-like activity in an acidic condition[4], decomposing $H_2O_2$ into highly cytotoxic hydroxyl radicals (•OH), thus augmenting the efficacy of tumor therapy. Additionally, due to the inherent localized surface plasmon resonance (SPR) effect of Pd-based nanomaterials[38,39], PMNS possesses intense near-infrared (NIR) absorption in both first NIR (NIR-I, 650–950 nm) and second NIR (NIR-II, 950–1700 nm) windows, endowing PMNS with excellent photothermal performance to afford the enhancement of catalytic activity and PA imaging ability. As a result, by means of 3D multispectral PA imaging, we achieve synchronously non-invasive measurement of local dynamic information of OxyHb, DeoxyHb and PMNS in real-time, and establishing underlying correlation between among those three molecular events to understand this catalytic process in vivo (Fig. 1). Therefore, the marriage of GOx and PMNS enables non-invasively 3D PA tracking of dynamically molecular events associated with cascade catalytic process, thus providing feedback for precise tumor catalytic therapy.

## Results

**Preparation and characterization in vitro performances of PMNS and PMNSG.** The PdMo bimetallene nanosheet (PMNS) was prepared following a one-pot wet-chemical approach[36]. The PMNS was dominated by sheet-like morphology with an average diameter of 56 nm and well characterized by transmission electron microscopy (TEM), high-resolution transmission electron microscopy (HRTEM), powder X-ray diffraction (XRD) and X-ray photoelectron spectroscopy (XPS) (Supplementary Figs. 1–4 and Fig. 2a). The PMNS was further modified with trithiol-terminated poly(methacrylic acid) (PTMP-PMAA), confirmed by fourier transform infrared (FT-IR) spectra (Supplementary Fig. 5). The ultraviolet-visible-near infrared (UV-Vis-NIR) spectra (Fig. 2b) indicated that PMNS possesses strong absorption in both NIR-I and NIR-II windows, following by concentration-dependent way. The extinction coefficients at 808 and 1064 nm were calculated to be 21.32 and 19.43 L g$^{-1}$ cm$^{-1}$, respectively (Fig. 2c), which are much higher than those of other reported nanomaterials[40,41]. Upon 1064 nm laser irradiation, the temperature of PMNS aqueous solution significantly rose in both concentration-dependent and laser power density-dependent manners (Fig. 2d and Supplementary Fig. 6a–c). Moreover, PMNS had superior photothermal stability (Supplementary Fig. 6d and Supplementary Fig. 7). The NIR-II photothermal conversion efficiency (PCE) of PMNS was calculated to be 60.4% (Supplementary Fig. 6e), which is remarkably higher than many other NIR-II photothermal agents[42,43]. The as-prepared PMNS also exhibited similar photothermal properties in NIR-I window (Supplementary Fig. 8). These results suggested PMNS hold great potential as both NIR-I and NIR-II PA imaging contrast agent.

Benefiting from the effective oxidoreductase activities of Pd-based nanomaterials[44], we then tested the CAT-like activity of PMNS. The time-dependent dissolved oxygen (DO) evolution process was monitored by the addition of $H_2O_2$ to PMNS solution. This process can be significantly enhanced by 1064 nm laser irradiation at a relatively low power density of 0.4 W cm$^{-2}$ (Fig. 2e). The $H_2O_2$ consumption studies also revealed that PMNS showed light-enhanced CAT-like activity (Supplementary Fig. 9). Next, we evaluated the POD-like activity (acid condition) of PMNS by recording the absorption changes in the substrate 3,3,5,5′-tetramethylbenzidine (TMB) at 652 nm (Supplementary Fig. 10)[45]. The TMB colorimetric reaction showed typical Michaelis–Menten kinetics with a $K_m$ value and a maximal reaction velocity ($V_{max}$) of 3.02 mM and 0.14 µM S$^{-1}$, respectively (Fig. 2f). Significantly, there was no color changed in

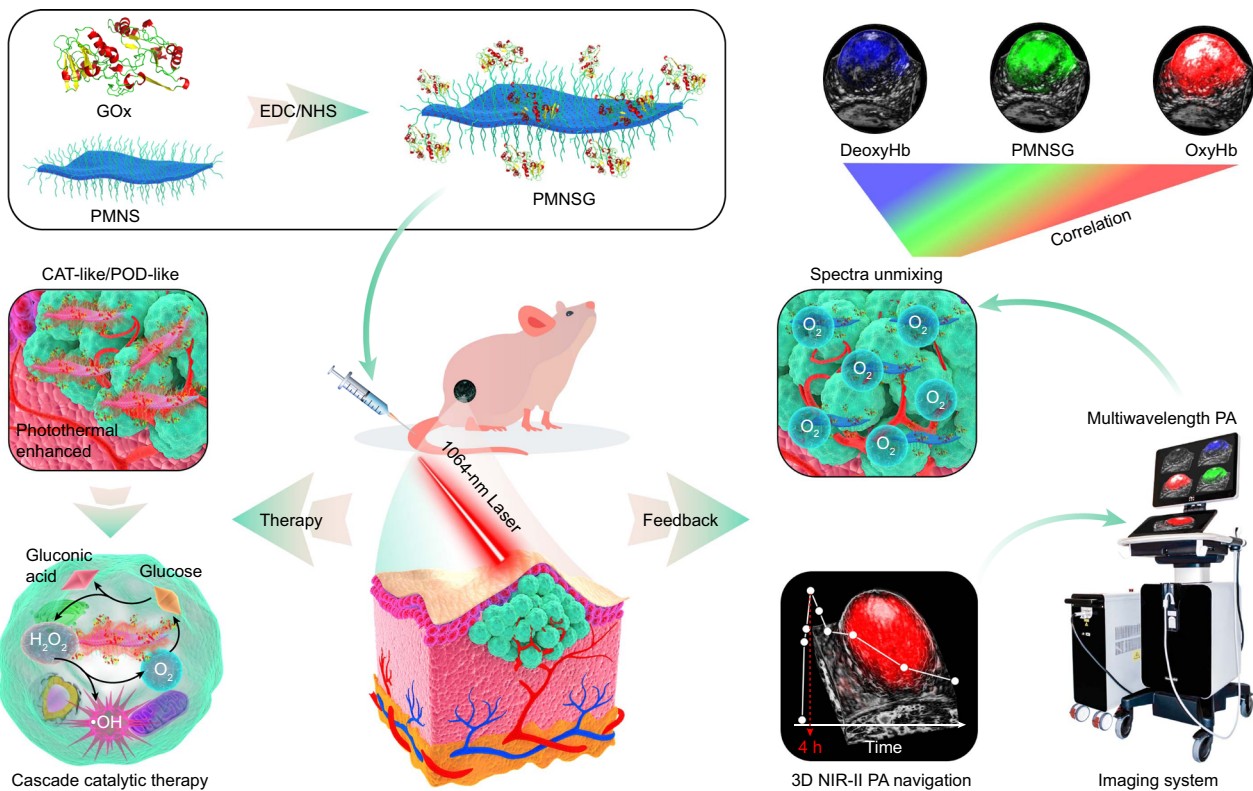

**Fig. 1 Preparation of cyclic cascade catalytic system of PMNSG for cancer treatment under the navigation of 3D PA imaging.** Schematic illustration of the preparation of PMNSG and its application in 3D PA feedback-looped cascade catalytic therapy.

neutral TMB solution with addition of PMNS plus $H_2O_2$ (Supplementary Fig. 10c), demonstrating the POD-like enzymatic reaction of PMNS does not proceed in this condition. In addition, the TMB oxidation under 1064 nm laser irradiation (3 min, 0.4 W cm$^{-2}$) showed a 1.96-fold enhancement compared to that of control group (Fig. 2g). According to the previous reports[46], transition metal doping and alloying effect can enhance the catalytic activity of Pd. Obviously, the strain fields of $\varepsilon_{xx}$, $\varepsilon_{yy}$ and $\varepsilon_{xy}$ obtained by geometric phase analysis (GPA) revealed that the PMNS exhibited wide range strain distribution along with different axes contributing to the enhanced activity (Fig. 2h)[47]. The corresponding linear strain scans (white arrow) obtained from Fig. 1h exhibited equivalent compressive strain and tensile strain (Fig. 2i), indicating catalyst uniformity to convincingly support the excellent catalytic activity of PMNS.

However, $H_2O_2$ is continuously consumed during PMNS-induced catalytic process. Thus, we introduced GOx to construct a cyclic cascade system (PMNSG). The GOx was conjugated on the surface of PMNS via an amidation reaction (Supplementary Fig. 5 and Supplementary Fig. 11), which does not affect the morphology of PMNS, the enzyme activity of GOx (Supplementary Fig. 12) and their dispersion stability (Supplementary Figs. 13-14 and Supplementary Table 4). GOx can efficiently catalyze the oxidation of glucose into gluconic acid and $H_2O_2$. The catalytic rate of GOx significantly depended on the $O_2$ concentration as such reaction was inhibited in anaerobic condition (Fig. 2j). Once the addition of $H_2O_2$, the pH value of $N_2$-saturated glucose/PMNSG solution decreased, which is contributed to the generation of gluconic acid, initiating by the new-generation of $O_2$ based on PMNS-mediated homolysis of $H_2O_2$. Meanwhile, the GOx-instructed $H_2O_2$ generation drove the whole catalytic process until the exhaustion of glucose. The above catalytic process could be also improved with 1064 nm laser irradiation (0.4 W cm$^{-2}$). Furthermore, in an acidic environment,

the $H_2O_2$ was converted into •OH by the POD-like PMNS, as shown in Supplementary Fig. 15. The TMB oxidation was inhibited under $N_2$-saturated condition, restarted with the addition of $H_2O_2$ and enhanced by the 1064-nm laser irradiation.

**In vitro photothermal-enhanced cascade catalytic therapy via NIR-II laser irradiation.** PMNSG exhibited low cytotoxicity on normal cells in vitro (Supplementary Fig. 16). The time-dependent red fluorescence enhancement and the corresponding mean fluorescence intensity analysis (Supplementary Figs. 17 and 18) demonstrated sufficient uptake of PMNSG in 4T1 cells at 4 h incubation. Under normoxia, after treatment of PMNSG (20 μg mL$^{-1}$), the 4T1 cell viability decreased to 22.0% (Fig. 3a). PMNS alone or free GOx showed mild toxicity (19.3% and 37.7%, respectively) (Supplementary Fig. 19a). Upon 1064 nm laser irradiation (5 min, 0.4 W cm$^{-2}$), the PMNS and PMNSG groups exhibited obvious cytotoxicity. Meanwhile, the therapeutic effect of PMNSG was also evaluated in hypoxia ($N_2/CO_2/O_2$: 94/5/1 in volume ratio). As shown in Fig. 3b, the PMNSG treatment showed remarkable cytotoxicity with laser irradiation enhancement. Free GOx induced little toxicity (~2.2%) under the hypoxic condition (Supplementary Fig. 19b). Meanwhile, the live/dead assay results further confirmed that PMNSG plus laser treatment could induce reproducible cell death (Supplementary Fig. 20). These results indicated that the PMNSG, $H_2O_2$ and $O_2$ were crucial for efficiently therapeutic outcome during NIR-II light-enhanced cyclic cascade reaction. Afterwards, cell membrane integrity was evaluated by lactase dehydrogenase (LDH) quantification assay (Fig. 3c)[48]. The PMNSG plus 1064-nm laser irradiation induced abundant leakage of LDH, up to 1.5-fold compared to control. Additionally, glucose played a critical role and could be depleted to block glycolysis, thus reducing the intracellular adenosine-5'-triphosphate (ATP)

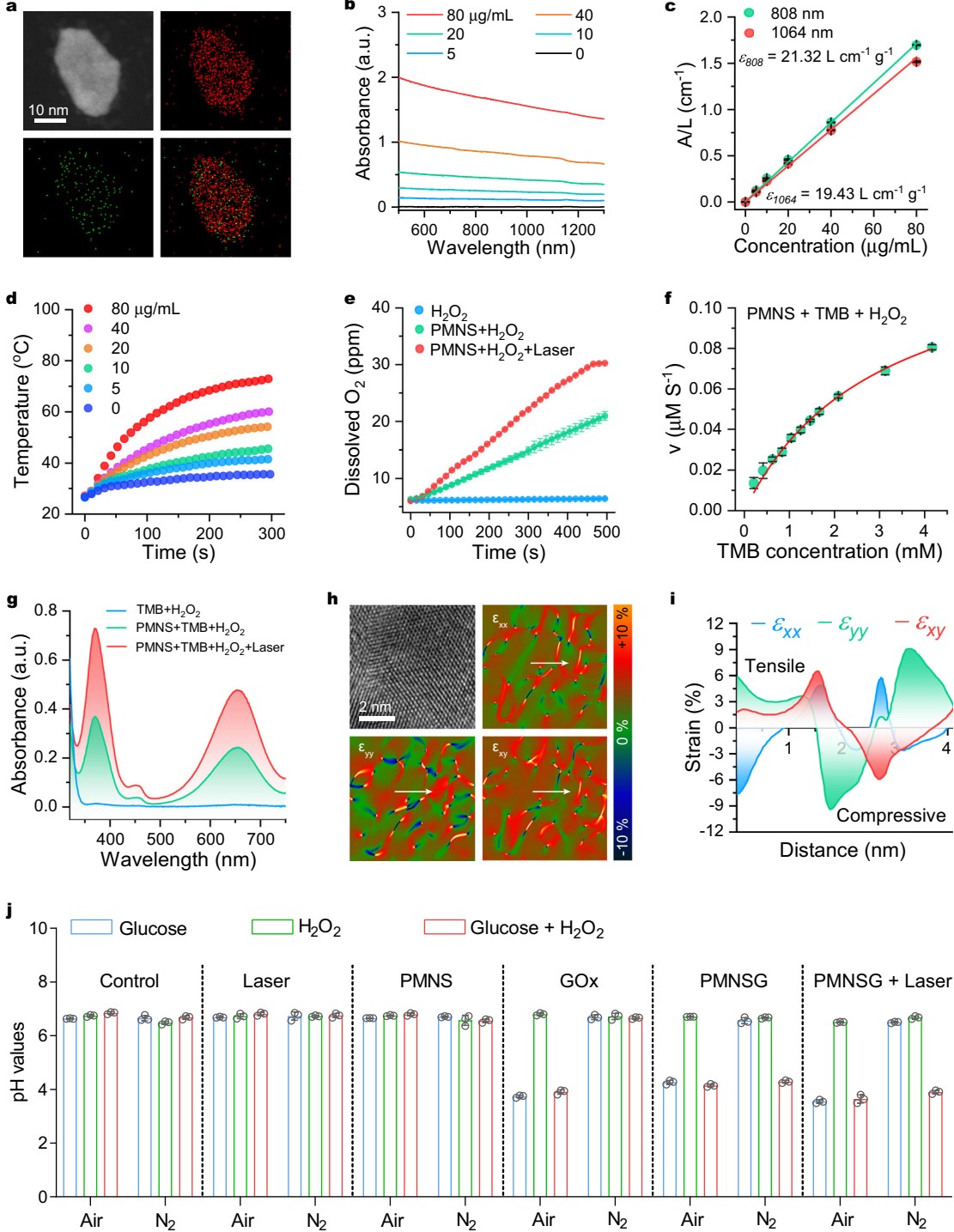

**Fig. 2 Characterization of PMNS and PMNSG. a** EDS elemental mapping of Pd and Mo in a single bimetallene nanosheet. **b** UV-Vis-NIR spectra of PMNS at different concentrations. **c** The relationships between the ratio of the absorbance of PMNS at 808 and 1064 nm to the length of cuvette (L) (A/L) on the PMNS concentration (C). Data are presented as mean ± SD. ($n = 3$). **d** Concentration-dependent photothermal curves of PMNS (0, 5, 10, 20, 40, 80 μg/ mL) under 1064-nm laser irradiation for 5 min. **e** $O_2$ generation in different treatments ($H_2O_2$, $H_2O_2$ + PMNS and $H_2O_2$ + PMNS + laser). Data are presented as mean ± SD. ($n = 3$). **f** Kinetics for POD-like activity of PMNS. Data are presented as mean ± SD. ($n = 3$). **g** UV-Vis absorption spectra of TMB in acid buffer solution (pH = 5.4) with and without 1064-nm laser irradiation (0.4 W cm$^{-2}$). **h** Surface strain mapping for PMNS along with different directions ($\varepsilon_{xx}$, $\varepsilon_{yy}$ and $\varepsilon_{xy}$), and **i** the corresponding strain distribution along the white arrows. **j** Variation of pH values after various treatment. Data are presented as mean ± SD. ($n = 3$). Source data are provided as a Source Data file.

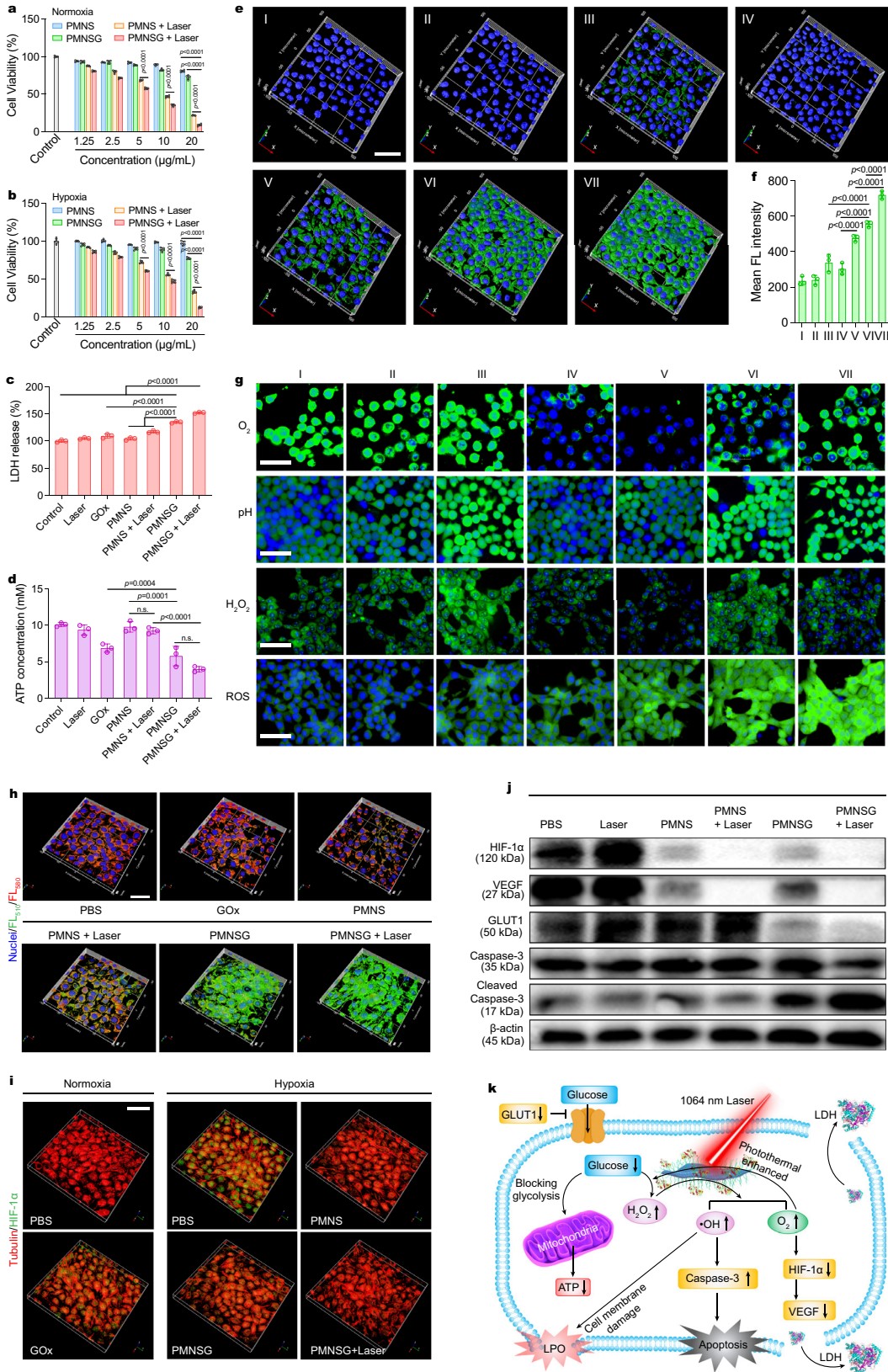

supply[49]. As shown in Fig. 3d, the PMNSG plus laser treatment induced the lowest ATP levels in tumor cells, indicating the enhanced cascade catalytic process could efficiently exhaust intracellular glucose.

To elucidate the cyclic cascade effect of PMNSG induced cytotoxicity, 4T1 cells in different treatment groups were co-stained with annexin V-FITC and propidium iodide (PI) (Supplementary Fig. 21) and analyzed with flow cytometry. PMNSG plus laser group induced the highest apotosis rate of 84.8%. Meanwhile, the caspase-3 activation (green fluorescence signal) was examined[50]. As shown in Fig. 3e, f, the PMNS alone and control groups exhibited low fluorescence signal.

**Fig. 3 In vitro photothermal-enhanced cascade catalytic therapy via mild NIR-II laser irradiation. a, b** In vitro different treatments (PBS, PMNS, PMNS + laser, PMNSG, and PMNSG + laser) of 4T1 cells in normoxia (**a**) and hypoxia (**b**) environments. Data are presented as mean values ± SD ($n = 5$). Statistical significance was calculated via one-way ANOVA with Tukey's multiple comparisons test. **c** Quantitative analysis of LDH release and **d** ATP assay of 4T1 cells after viaious treatments. Data are presented as mean ± SD ($n = 3$). Statistical significance was calculated via one-way ANOVA with Tukey's multiple comparisons test. n.s. no significance. **e** Activation of caspase-3 in 4T1 cells after various treatments for 24 h (I: PBS; II: Laser; III: GOx; IV: PMNS; V: PMNS + laser; VI: PMNSG; VII: PMNSG + laser). **f** Quantification of green fluorescence signals from caspase-3 under different treating conditions. Data are presented as mean ± SD. ($n = 3$). Statistical significance was calculated via one-way ANOVA with Tukey's multiple comparisons test. **g** High content images of 4T1 cells stained with RDPP, BCECF-AM, Amplex Red and DCFH-DA after different treatments. Scale bar is 50 μm. $n = 5$ independent experiments. **h** Evaluation of LPO in 4T1 cells after different treatments. The LPO was determined by the fluorescence intensity ratio of $FL_{510}/FL_{580}$ (green fluorescence/red fluorescence). Scale bar is 50 μm. **i** Immunofluorescence imaging of 4T1 cells with stained anti-α-tubulin (red) and anti-HIF-1α (green) after different treatments under normoxic and hypoxic conditions (1% $O_2$, 5%$CO_2$, and 94% $N_2$), respectively. Scale bar is 50 μm. **j** Western immunoblots analysis of expression levels of cascade catalytic therapy related proteins (HIF-1α, VEGF, GLUT1 and caspase-3) in 4T1 cells. $n = 3$ independent experiments. **k** Possible mechanism of PMNSG plus laser-mediated catalytic therapy. Source data are provided as a Source Data file.

For free GOx group, a weak green fluorescence signal can be observed. Obviously, PMNSG induced significant expression of caspase-3, which could be further enhanced by laser irradiation. These results indicated that cell death was mainly attributed to the caspase-3 activation induced apotosis.

To further understand the mechanism of caspase-3 mediated cell death, we studied the variation of intracellular molecular events associated with the cascade reaction. First, after incubation with PMNSG, the cancer cells showed higher ROS levels than those of normal cells (Supplementary Fig. 22), demonstrating the specificity of PMNSG toward cancer cells. 4T1 cells treated with PMNS showed a decrease of green fluorescence intensity ($[Ru(dpp)_3]Cl_2$, RDPP, a $O_2$ probe) (Fig. 3g and Supplementary Fig. 23a), which further decreased with NIR-II laser irradiation enhanced CAT-like activity of PMNS. Besides, the PMNSG treated group resulted higher signals of BCECF-AM (pH probe), hydroxyphenyl fluorescein (HPF, •OH probe)[51] and 2',7'-dichlorodihydrofluorescein diacetate (DCFH-DA, ROS probe) but lower signal of Amplex Red ($H_2O_2$ probe) than those of free GOx treated group (Fig. 3g, Supplementary Fig. 23b–d and Supplementary Figs. 24–27), strongly suggesting the operation of cascade catalytic reaction that could be further reinfored by laser irradiation. Of note, the efficient •OH generation can damage cell membrane due to its induction of lipid peroxidation (LPO)[52]. As a result, much stronger green fluorescence was observed in PMNSG-treated cells than the control group, and further enhanced by 1064-nm laser irradiation (0.4 W cm$^{-2}$), suggesting that the endocytosed PMNSG triggered significant LPO in 4T1 cells during photothermal-enhanced cascade catalytic process (Fig. 3h and Supplementary Fig. 28). In addition, the generated $O_2$ can alleviate the hypoxic condition, resulting in the down-regulated expression of HIF-1α protein in cancer cells[53]. As shown in Fig. 3i, green immunofluorescence signals of HIF-1α protein can only be observed under hypoxic condition. Upon incubation with PMNS, the expression of HIF-1α protein was significantly down-regulated, and enhanced by laser irradiation (Supplementary Fig. 29). Importantly, for PMNSG treated group, even the GOx catalytic process comsumed $O_2$, the expression of HIF-1α protein was also down-regulated, indicating the effective $O_2$ generation ability of PMNS-based CAT-like activity. Given that HIF-1α could induce the transactivation a variety of signal molecules, including vascular endothelial growth factor (VEGF) to increase angiogenesis and promote tumor growth[54], we applied Western blot to investigate the expression of HIF-1α and VEGF under different conditions (Fig. 3j). As a result, the PMNSG plus laser treatment group significantly induced the down-regulation of HIF-1α and VEGF and exhibited a similar tendency as compared with immunofluorescence staining images (Fig. 3i), while negetively correlated with $O_2$ generation. Moreover, the efficient $O_2$ generation was beneficial for GOx-mediated aerobic

catalysis toward glucose, thus causing severe glucose depletion to reduce the intracellular adenosine-5'-triphosphate (ATP) supply (Fig. 3d). GLUT1, a glucose transporter protein, primarily regulates the cell uptake of glucose[55]. The GOx-mediated aerobic catalysis could block the glycolysis in cells, down-regulating the GLUT1 expression to further decrease the cell uptake of glucose[56]. Of note, the PMNSG showed higher negative regulation effect on GLUT1 expression, which could be further significantly down-regulated by laser irradiation (Fig. 3j). Benefiting from the efficient cascade catalytic process, PMNSG can induce significant cell apoptosis with laser irradiation, as demonstrated by the obviously upregulated caspase-3 expression (Fig. 3j). These results suggested that the above cascade catalytic process in living cells accompanied by the occurrence of multimolecular events, thus providing a feedback and guidance for tumor therapy. Taken together, the summarized molecular mechanism of PMNSG plus laser-mediated catalytic therapy is shown in Fig. 3k. Our study identified that the combination of PMNS and GOx with 1064 nm laser irradiation significantly mediate the ATP level reduction, •OH level increase, HIF-1α/VEGF/GLUT1 down-regulation and caspase-3 activation, thus eficiently inducing the cell apoptosis.

**NIR-II 3D PA imaging tracking of PMNSG.** Benefiting from the high-resolution of NIR-II 3D PA imaging[33,34,57], the NIR-II 3D PA signals were obtained by the Vevo LAZR-X PA imaging system. To ensure the feasibility of in vivo PA imaging, we first evaluated the PA imaging ability of PMNSG and PMNS in physiological condition. As a result, the PA signals enhancement of PMNSG and PMNS occurred in a concentration-dependent manner (Fig. 4a, b). Even at a low concentration (15 μg mL$^{-1}$), the PMNSG or PMNS also possessed high PA amplitude. Afterwards, PA imaging was conducted in vivo. Using 3D-rendered images, we mapped the NIR-II PA signals in tumor tissues (Fig. 4c). After intravenous (i.v.) injection of PMNSG or PMNS, the NIR-II PA signals of tumor tissues increased and reached the maximum at 4 h, showing 7.7 and 4.4-fold enhancement compared to that of background, respectively (Fig. 4c, d). Intriguingly, the mice treated with PMNSG showed a 1.6-fold higher PA amplitude than that of treated with PMNS. The high tumor accumulation of PMNSG or PMNS may contribute to the passive targeting pathway of enhanced permeability and retention (EPR) effect[32]. Thereafter, the PA signals gradually decreased, probably due to the efflux of PMNSG or PMNS from tumor tissues. Utilizing 3D NIR-II PA imaging with high signal noise ratio (SNR) and efficient imaging depth, tomography images-depth resolved B-scan PA/US images were obtained. As shown in Fig. 4e, the NIR-II PA signals of PMNSG or PMNS illuminated the entire tumor region, indicating the uniform intratumoral distribution of PMNSG or PMNS. Moreover, the biodistribution of PMNSG or PMNS was measured by ex vivo PA imaging of organs

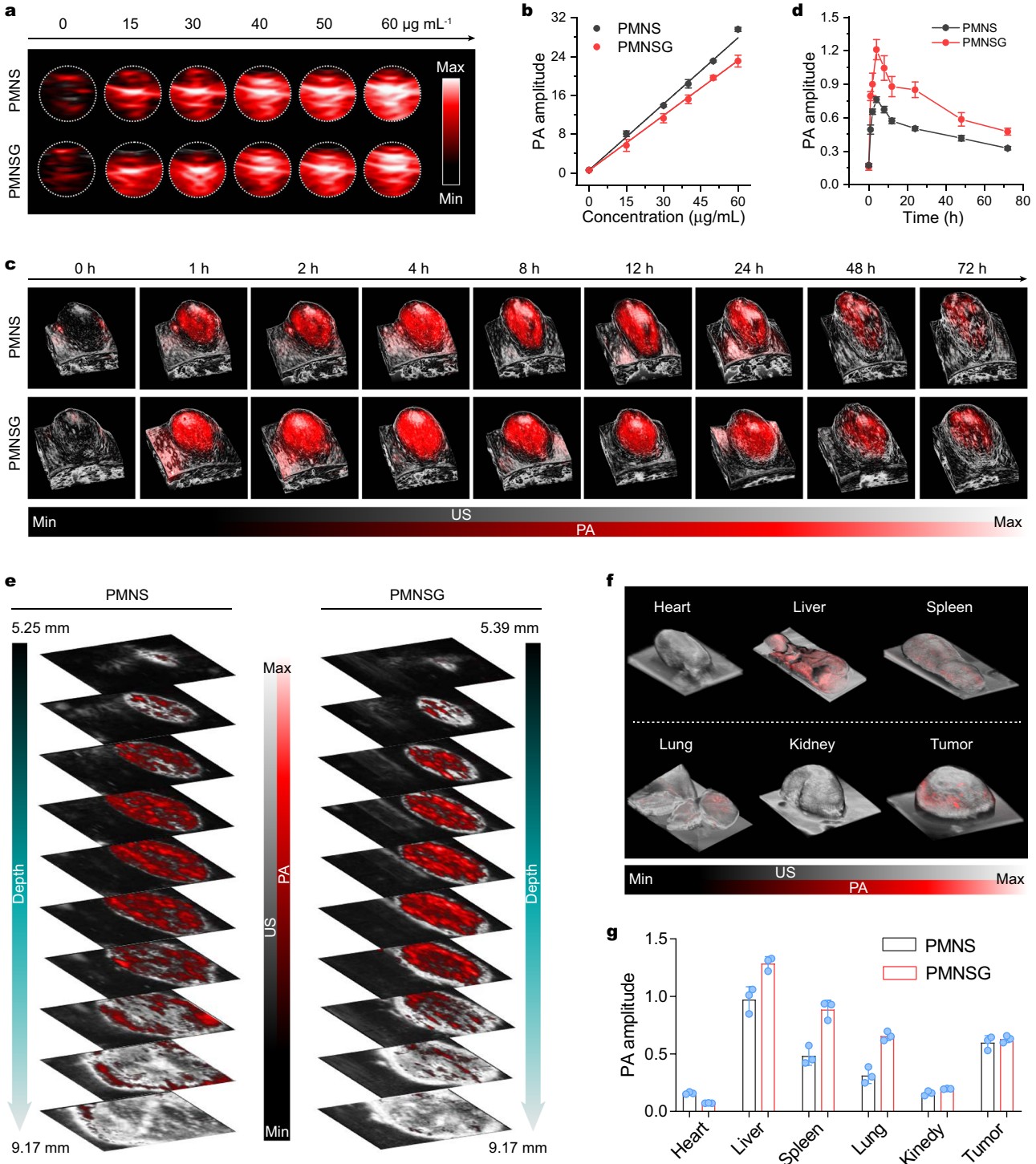

**Fig. 4 In vitro/vivo and ex vivo NIR-II PA imaging of PMNSG. a**, **b** In vitro NIR-II PA images of PMNS and PMNSG at different concentrations and the corresponding liner fits of NIR-II PA amplitude as a function of concentration at 1280 nm, respectively. **c**, **d** Time-dependent 3D-rendered NIR-II PA/US images of tumor sites after i.v. injection of PMNS and PMNSG and the corresponding quantification of PA amplitude as a function of time, respectively. Data are presented as mean ± SD. ($n = 3$). **e** Tomography images-depth resolved B-scan PA/US images of tumor regions after intravenous administration of PMNS and PMNSG. **f** Ex vivo 3D-rendered PA/US images of various organs. **g** Quantification of NIR-II PA amplitude in (**f**) (Data are presented as mean ± SD. $n = 3$). Source data are provided as a Source Data file.

at 24 h post-injection (Fig. 4f, g and Supplementary Fig. 30). These results revealed that the partial PMNSG or PMNS accumulated in liver, followed by another organs, which were consistent with the results obtained via inductively coupled plasma massspectrometry (ICP-MS) (Supplementary Fig. 31). The blood circulation of

PMNSG was also evaluated and exhibited a relatively long circulation time of 5.4 h (Supplementary Fig. 32). Besides, the PMNSG possessed good biosafety and biodegradability, confirmed by the hemolysis, histological examination and biodegradation behavior investigation (Supplementary Figs. 33–35).

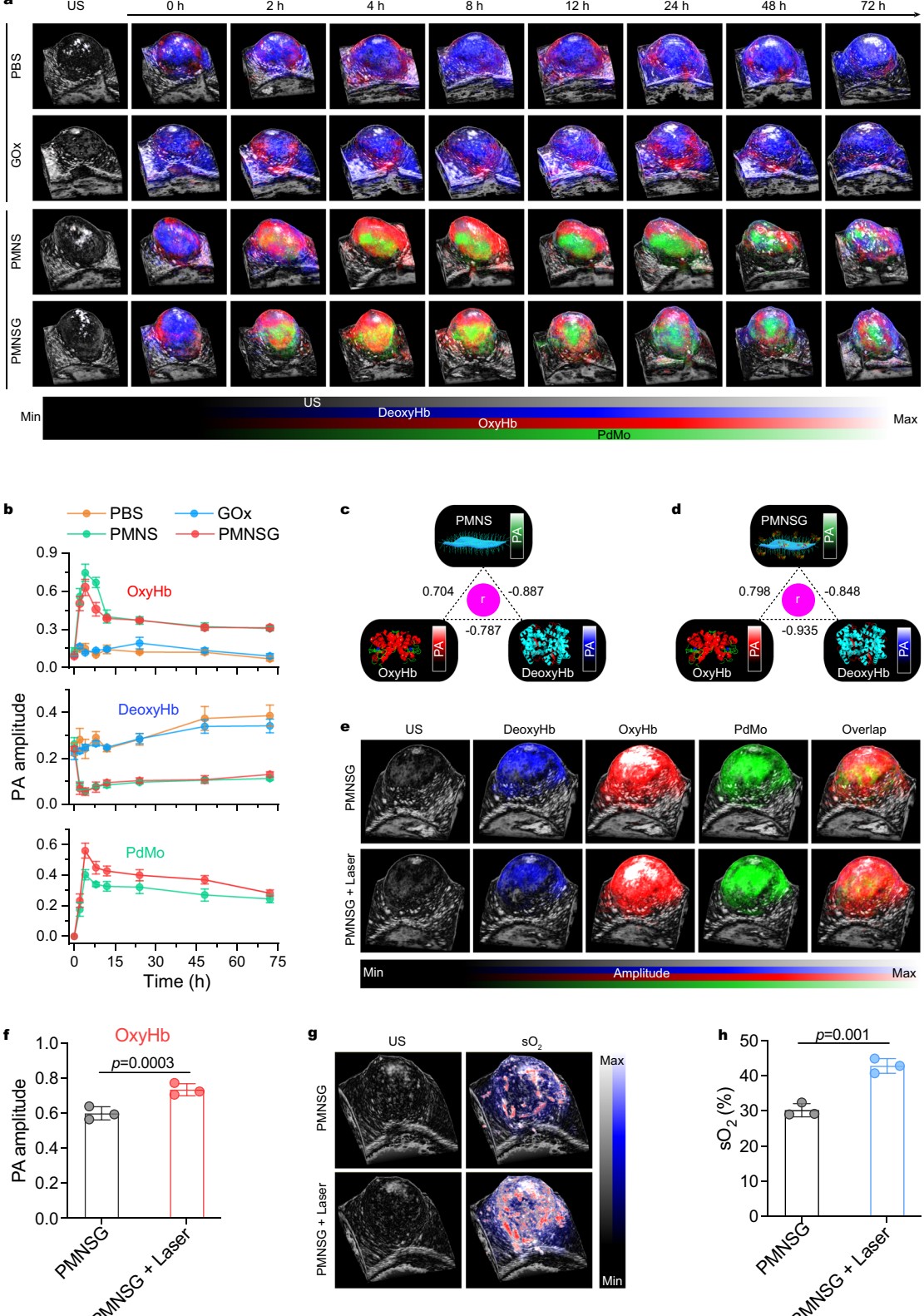

**In vivo synchronous 3D PA imaging monitoring in cascade catalytic process.** We then investigated the dynamic molecular events associated with the cascade catalytic process in vivo. Benefiting from the PA imaging responses of PdMo component, OxyHb, and DeoxyHb in the NIR-I region[26], multiwavelength (680–970 nm) PA imaging of tumors with different treatments was performed on 4T1 tumor-bearing mice (Supplementary Fig. 36). The time-course 3D-rendered B-mode PA/US images (Fig. 5a) of OxyHb, DeoxyHb and PdMo in tumors, and the corresponding quantitative values (Fig. 5b) obtained from spectral unmixing method suggested that the content variation of OxyHb and DeoxyHb in PMNS or PMNSG treated group

**Fig. 5 In vivo simultaneous PA imaging monitoring of OxyHb, DeoxyHb and PMNSG in real-time. a, b** Time-course of 3D-rendered B-mode PA/US images of OxyHb, DeoxyHb and PdMo in tumor region after i.v. injection of PBS, GOx, PMNS and PMNSG (**a**) and the corresponding quantification of PA values as a function of time, respectively (**b**). Data are presented as mean ± SD. (n = 3). **c, d** Correlation analysis among DeoxyHb, OxyHb and PMNS (**c**) or PMNSG (**d**) signal intensities shown in **b**. **e** 3D-rendered B-mode PA/US images of mice after PMNSG treatment for 4 h and then 1064-nm laser irradiation for 10 min. **f** Quantification of PA values for oxyHb in **e**. Data are presented as mean ± SD. (n = 3). Statistical significance was calculated via one-tailed Student's t test. **g** 3D-rendered B-mode PA/US images of sO$_2$ after PMNSG treatment for 4 h and then 1064-nm laser irradiation for 10 min. **h** Quantification of PA values for sO$_2$ values in (**g**). Data are presented as mean ± SD. (n = 3). Statistical significance was calculated via one-tailed Student's t test. Source data are provided as a Source Data file.

strongly depend on the accumulation of PMNS or PMNSG, when compared to that of PBS and free GOx treated groups. To further support this result, Pearson correlation was tested among the separated PA signals of OxyHb, DeoxyHb and PdMo[31]. As a result, the PMNS or PMNSG signals were positively correlated with OxyHb (r = 0.704 and 0.798, respectively) and negatively correlated with DeoxyHb (r = −0.887 and −0.848, respectively) (Fig. 5b–d). Besides, the time-course NIR-I PA signals variation of PMNS or PMNSG were strongly consistent with the NIR-II PA imaging (Fig. 4c, d) results. Therefore, these results convincingly confirmed that the efficient O$_2$ generation in tumor tissues mediated by the high tumor accumulation of PMNS or PMNSG. Concretely, the intratumoral blood oxygen saturation (sO$_2$) was further monitored quantitatively in real time by the multi-wavelength PA and B-Mode US imaging (Supplementary Fig. 37)[28]. These results demonstrated the efficient oxygen supply ability of PMNS for the aerobic catalysis of GOx toward glucose. In addition, the 3D-rendered B-mode PA/US images of OxyHb and sO$_2$ revealed that the 1064-nm laser irradiation could drive the cascade reaction process in vivo, and this enhancement effect covered the entire tumor region (Fig. 5e–h).

**In vivo NIR-II photothermal-enhanced cascade catalytic therapy under the navigation of PA imaging.** Subsequently, we investigated the in vivo therapeutic effect of PMNSG on 4T1 tumor-bearing mice after systemic administration (Fig. 6a). According to the above PA imaging results, the tumors were treated with a 1064-nm laser irradiation at 4 h post i.v. injection of PMNSG. Firstly, the in vivo photothermal performance of PMNSG under laser irradiation (0.4 W cm$^{-2}$, 10 min) was examined. Upon 3 min laser irradiation, the PMNS and PMNSG induced the tumor temperature changes of 7.4 and 8.4 °C, respectively, while that of control group only showed a temperature change of 3.6 °C (Fig. 6b, c). Thereafter, the temperature changes in the tumor region of mice treated with PMNS and PMNSG plus laser irradiation held steady at about 8.8 and 8.9 °C, respectively, until the end of 10 min irradiation. After different treatments, the tumor growth of mice was monitored by B-mode ultrasonography with the 3D reconstruction of tumor region (Supplementary Figs. 38–40 and Fig. 6d, e). For mice treated with PBS, Laser, GOx or PMNS alone, the tumor growth showed negligible inhibition effect. For the PMNSG treated group, the tumor growth was significantly suppressed, demonstrating that the effective catalytic therapeutic performance of PMNSG in vivo. In addition, the mice treated with PMNS plus 1064-nm laser irradiation only showed partial suppression of tumor growth, while the treatment of PMNSG plus laser irradiation exhibited most conspicuous inhibition effect on tumor growth, suggested good therapeutic efficacy of photothermal-enhanced catalytic therapy in vivo. Notably, the tumor volumes obtained from the 3D reconstruction were coincided with that of experimental measurements, demonstrating the feasibility of applying B-mode ultrasonography to monitor the evolution of tumor size in real-time. Finally, we collected the tumors and main organs from all groups at 14 days. The average weight of tumors from mice

treated with PMNSG plus 1064-nm laser irradiation was lowest (Fig. 6f and Supplementary Fig. 41). The body weights of mice after different treatments were monitored for two weeks, and showed no obvious variations (Fig. 6g).

In addition, the histological evaluation of tumor slices after 24 h PMNSG plus 1064 nm laser treatment for the hematoxylin-eosin (H&E), Ki-67 and terminal deoxynucleotidyl transferase-mediated deoxyuridine triphosphate nick end labeling (TUNEL) staining showed most significant tumor tissue damage, proliferation inhibition and cell apoptosis extent, respectively (Fig. 6h). At the same time, we observed the ROS level significantly increased at 4 h post injection of PMNSG, and which was elevated by the laser irradiation (Supplementary Fig. 42). In consideration of deep penetration of NIR-II light, we also evaluated the expression of hypoxia and apoptosis biomarker (HIF-1α and caspase-3, respectively) at the molecular level in different thickness of tumor tissue. Immunofluorescence staining of different thickness (2, 4, and 6 mm) of tumor tissue slices showed that tumors treated with PMNSG plus laser irradiation possessed significantly improvement of hypoxia reflected by HIF-1α expression and significant cell apoptosis reflected by caspase-3 expression even at 6 mm depth of tumor tissues (Fig. 6i–l and Supplementary Fig. 43). Of note, the blood biochemistry analysis results (Supplementary Fig. 44) indicated that PMNSG exhibited excellent biocompatibility. Meanwhile, no pathological abnormality or inflammation was observed in major organs for PMNSG (Supplementary Fig. 45), and no obvious signal of Ly6G$^+$ neutrophils was observed in liver slice of PMNSG treatment group (Supplementary Fig. 46)[58], indicating the long-term biosafety of PMNSG. Taken together, the as-prepared PMNSG could be an efficient and safe theranostic nanoagent for PA imaging-guided catalytic therapy.

## Discussion

In summary, we constructed a biodegradable cyclic cascade catalytic reaction system (PMNSG) via the combination of PMNS and GOx for cascade catalytic therapy. After intravenous injection of PMNSG, the intracellular H$_2$O$_2$ level was elevated by the aerobic catalysis of GOx, promoting the nanozyme-mediated catalytic therapy while starving the tumor. Meanwhile, the PMNS not only exhibits CAT-like activity to catalyze the decomposition of H$_2$O$_2$ into O$_2$, inducing the PA signals variation of OxyHb and DeoxyHb, potentiating aerobic catalysis of GOx toward glucose and reducing the systemic toxicity of GOx, but also shows POD-like activity to decompose H$_2$O$_2$ into highly cytotoxic •OH. Therefore, our cascade catalytic reaction system can significantly induce the dynamic occurrence of multi-molecular events while augmenting the efficacy of tumor therapy. Moreover, the PMNS possesses excellent NIR-II photothermal performance, allowing to strengthen the above cyclic cascade catalytic process and provide the PA navigation. As a result, we can achieve synchronously non-invasive monitoring of dynamic molecular events during cascade catalytic process, providing a correlation between endogenous and exogenous molecules to deepen our understanding of this catalytic process in vivo, thus advancing the exploration of nanozyme-based biomedical applications. Therefore, our study

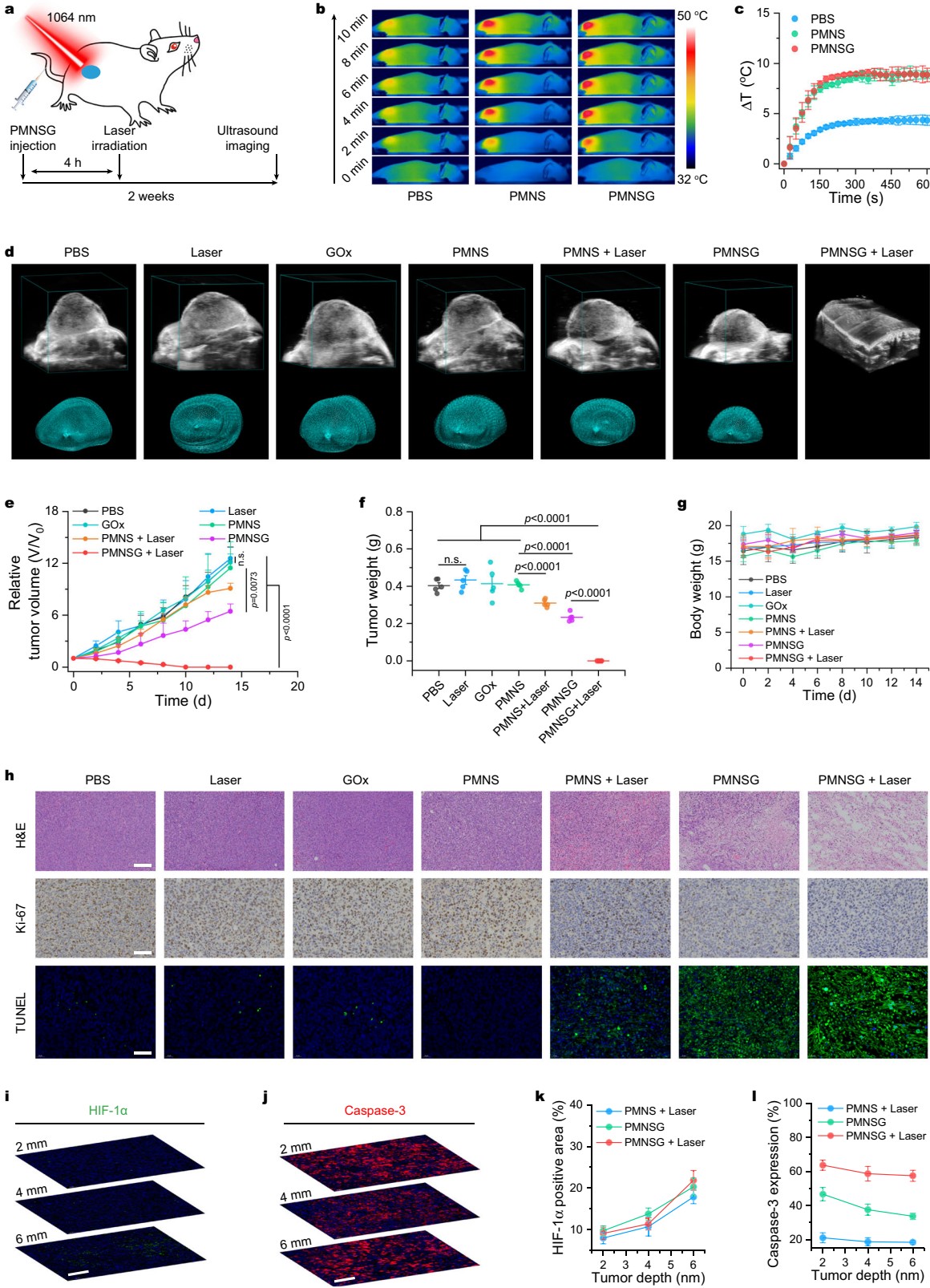

highlights the potential of 3D PA imaging in feedback-based cascade catalytic therapy.

## Methods

**Materials**. N-hydroxysuccinimide (NHS), H$_2$O$_2$ (30%), 3,3',5,5'-tetra-methylbenzidine (TMB) and 1-ethyl-3-(3-dimethyl aminopropyl) carbodiimide (EDC) were purchased from Sigma-Aldrich Co., Ltd. 3-(4,5-Dimethylthiazol-2-yl)-2,5-diphenyltetrazolium bromide (MTT), calcein AM, propidium iodide (PI), reactive oxygen species (ROS) assay kit (2',7'-dichlorofluorescin diacetate, DCFH-DA), 4',6'-diamidino-2-phenylidole (DAPI), hydroxyphenyl fluorescein (HPF), RDPP, BCECF-AM, Annexin V-FITC/PI Apoptosis Assay Kit, ATP assay kit, LDH cytotoxicity assay kit and penicillin-streptomycin solution (100X) were purchased from Beyotime Institute of Biotechnology (Shanghai, China). Glucose oxidase (GOx, 100000-250000 units g$^{-1}$) and anti-α-Tubulin (catalogue no. T9026) were purchased from Sigma-Aldrich Co., Ltd. C11 BODIPY 581/591 was purchased

**Fig. 6 In vivo NIR-II photothermal-enhanced catalytic therapy via PMNSG. a** Schematic of therapeutic approach for tumor-bearing mice. **b, c** IR thermal images (**b**) and the corresponding tumor temperature change (**c**) of mice after different treatments. Data are presented as mean ± SD ($n = 3$). **d** B-mode US images with the 3D reconstruction of tumor region for different treatment groups. **e** Tumor growth curves of all groups after various treatments. Data are presented as mean ± SD ($n = 5$). Statistical significance was calculated with two-tailed Student's $t$ test. **f, g** Relative tumor weight (**f**) and average body weight (**g**) of 4T-1 tumor-bearing mice after different treatments. Data are presented as mean ± SD ($n = 5$). Statistical significance was calculated with two-tailed Student's $t$ test. **h** H&E, Ki-67 and TUNEL staining of tumor slices collected from different groups after various treatments indicated. Scale bar is 100 μm. **i, j** Immunofluorescence staining images of tumor slices at different laser irradiation depth after treated with PMNSG plus 1064 nm laser (HIF-1α (**i**) and caspase-3 (**j**)). **k, l** Quantification of HIF-1α expression (**k**), and caspase-3 expression (**l**) of tumor at different laser irradiation depth after various treatments. Data are presented as mean ± SD ($n = 3$). Source data are provided as a Source Data file.

from Cayman Chemical. Anti-HIF-1α (catalogue no. 36169 T), anti-caspase-3 (catalogue no. 36169 T), anti-HIF-1α (catalogue no. 14179 S), anti-caspase-3 (catalogue no. 9662 S), anti-cleaved caspase-3 (catalogue no. 9661 S, Cell Signaling Technology), anti-GLUT1 (catalogue no. 12939 S) and β-actin (catalogue no. 8457 S) were purchased from Cell Signaling Technology. Anti-VEGF (catalogue no. ab214424) was purchased from Abcam. Anti-Ki-67 (catalogue no. GB111499), anti-Ly6G (catalogue no. GB112299) and TUNEL (G1510) were purchased from Servicebio. Fetal bovine serum (FBS) and Dulbecco's modified Eagle's medium (DMEM) were purchased from GIBCO Co., Ltd. All cell lines (4T1, MCF-7, A375, 293 T, B6F10, Huh7 and HeLa) were purchased from Cell Bank of Chinese Academy of Sciences (Shanghai, China).

**Synthesis of PMNS**. The PdMo bimetallenes were prepared by a one-pot wet-chemical approach[36]. Next, the oleylamine stabilized PdMo bimetallenes were washed with ethanol three times and then dispersed in ethanol (5 mL), followed by the addition of poly(methacrylic acid) functionalized with pentaerythritol tetrakis 3-mercaptopropionate (PTMP-PMAA) (50 mg) with continuously stirring. After stirring for 12 h at room temperature, the final products were collected by centrifugation and washed with water for three times. Finally, the PTMP-PMAA modified Pd bimetallenes (PMNS) were dispersed in PBS (5 mL, pH 7.4) for further use.

**Synthesis of PMNSG**. GOx was conjugated on the surface of PMNS by the amidation reaction. Briefly, the 1-(3-Dimethylaminopropyl)-3-ethylcarbodiimide hydrochloride (EDC, 10 mg, 0.065 mmol) and N-hydroxysuccinimide (NHS, 5 mg, 0.043 mmol) were added into 5 mL of PMNS dispersion (1 mg mL$^{-1}$) under magnetic stirring for 2 h. After that, 40 μL of GOx (2 mg mL$^{-1}$) was added into the mixture and stirred for another 6 h. Finally, the PMNSG was collected by centrifugation and washed with water three times, and then dispersed in 5 mL of PBS (pH 7.4) for further use.

**Photothermal performance of PMNS**. The photothermal performance of PMNS was evaluated in eppendorf tube (200 μL) containing different concentration of PMNS dispersion. The photothermal stability of PMNS was performed by NIR laser (808 or 1064 nm) irradiation for five laser on/off cycles. The photothermal conversion efficiency of the PMNS was calculated as detailed in Supplementary Information.

**Catalase-like activity assay of PMNS**. Catalase (CAT)-like activity of PMNS were carried out by the production of O$_2$ in the presence of H$_2$O$_2$ at room temperature. Typically, 200 μL of PMNS (20 μg mL$^{-1}$) was added to 3 mL of PBS (pH 7.4, 0.1 M), at the same time, 200 μL of 30% H$_2$O$_2$ was added into the mixture. For photothermal enhanced CAT-like activity study, the mixture was continuously irradiated by 1064-nm laser with a power density of 0.4 W cm$^{-2}$. The concentration of generated O$_2$ were recorded every 15 s by using a specific oxygen electrode on Multi-Parameter Analyzer (JPSJ-605F, Leici China).

**Peroxidase-like activity assay of PMNS**. The peroxidase (POD)-like activity of PMNS was evaluated by using TMB as the substrate in the presence of H$_2$O$_2$ in HAc-NaAc buffer solution (0.1 M, pH 4.5). Typically, 10 μL TMB (in DMSO, 10 mg mL$^{-1}$) and 10 μL PMNS solution (16 μg mL$^{-1}$) were added into 2 mL of HAc-NaAc buffer solution, followed by the addition of 10 μL 30% H$_2$O$_2$. The steady-state kinetic assays were measured at 37 °C. The kinetic assays of PMNS with H$_2$O$_2$ as the substrate were conducted by adding 10 μL of 10 mg mL$^{-1}$ TMB and different amounts (10, 20, 30, 40, 50, 60, 70, 80, 100, 150, 200 μL) of 30% H$_2$O$_2$ solution. The kinetic assays of PMNS with TMB as the substrate were conducted by adding 10 μL 30% H$_2$O$_2$ and different amounts (10, 20, 30, 40, 50, 60, 70, 80, 100, 150, 200 μL) of TMB solution (10 mg mL$^{-1}$). All of the above reactions were performed by measuring the absorbance of TMB (652 nm) at different reaction time points, and the Michaelis–Menten constant was calculated according to the Michaelis–Menten saturation curve. For photothermal enhanced POD-like activity study, the mixture was irradiated by 1064-nm laser with a power density of 0.4 W cm$^{-2}$. For control group, the POD-like activity of PMNS was evaluated in PBS (0.1 M, pH 7.4).

**Photothermal-enhanced cascade catalytic reaction**. For the variation of pH value, 100 μL of H$_2$O, PMNS (1 mg mL$^{-1}$), GOx (2 μg mL$^{-1}$) or PMNSG (1 mg mL$^{-1}$) solution were added into 2 mL of glucose solutions (2 mg mL$^{-1}$ in water, pH 6.8) with or without addition of 30% H$_2$O$_2$ (10 μL) at 37 °C under normoxic or hypoxic condition (pre-saturated with N$_2$) with or without 1064-nm laser irradiation (0.4 W cm$^{-2}$). The variation of pH values was recorded at different times via a pH meter. For the investigation of enhanced POD-like activity, 10 μL TMB (in DMSO, 10 mg mL$^{-1}$) and 10 μL PMNS solution (16 μg mL$^{-1}$) were added into 2 mL of HAc-NaAc buffer solution with or without 10 μL 30% H$_2$O$_2$ under normoxic or hypoxic condition with or without 1064-nm laser irradiation (0.4 W cm$^{-2}$).

**In vitro cancer therapy**. For the cytotoxicity evaluation, 4T1 cells were cultured in a 96-well plate at a density of $1 \times 10^4$ cells/well at 37 °C with 5% CO$_2$ humidified atmosphere for 24 h. Subsequently, the culture medium was removed. The cells were incubated with fresh medium containing PMNS or PMNSG with different concentrations for another 24 or 48 h, and then washed with DMEM two times. The cell viabilities were evaluated by 3-(4,5-Dimethylthiazol-2-yl)-2,5diphenylte-trazolium bromide (MTT) assay. For in vitro photothermal-enhanced cascade catalytic therapy, the 4T1 cells were incubated with different concentration of PMNSG under normoxic or hypoxic condition (N$_2$/CO$_2$/O$_2$: 94/5/1 in volume ratio), followed by the 1064-nm laser irradiation (0.4 W cm$^{-2}$) for 5 min. For other groups (control, free GOx, PMNS, PMNS + laser and PMNSG), the cells were conducted various treatments. Finally, the cell viabilities were evaluated by MTT assay.

**LDH assay**. Cell membrane integrity was evaluated by measuring LDH activity. After the 4T1 cells were conducted different treatments, the supernatants were collected, and the amount of LDH released from cells was determined by LDH cytotoxicity assay kit (Beyotime Biotechnology, China). The absorbance was measured by microplate reader at 490 nm. The data in each treatment group is expressed as a percentage of control (PBS treatment).

**Live/dead staining assay**. 4T1 cells were cultured in a 48-well plate at 37 °C under hypoxic condition for 24 h. Subsequently, the culture medium was removed. The cells were incubated with fresh medium containing PBS, GOx, PMNS and PMNSG for 4 h, followed by treating with or without 1064-nm laser irradiation (0.4 W cm$^{-2}$) for 5 min, and further cultured for another 4 h. After that, the cells were co-stained with calcein AM and PI for 0.5 h, then washed with fresh medium two times and imaged by using a fluorescence inverted microscope (Olympus UHGLGPS, China).

**Caspase-3 assay**. 4T1 cells were cultured in 96-well plate. After various treatments, cells were fixed with 1% paraformaldehyde at 37 °C for 10 min and permeabilized with PBS containing 0.2% Triton X-100 at 37 °C for 10 min. Next, the blocking step was conducted by PBS with 0.05% Tween-20 buffer (containing 1% bovine serum albumin) at room temperature for 30 min. After that, the cells were incubated with anti-caspase-3 (1:1000, catalogue no. 36169 T, Cell Signaling Technology) for 12 h, followed by labeling with secondary antibodies (Alexa Fluor 488-conjugated AffiniPure Goat Anti-Rabbit IgG (H + L)) at room temperature for 1 h. Subsequently, the cells were stained with DAPI for 10 min. Finally, the fluorescence intensities were analyzed by High Content System (PerkinElmer).

**ROS and •OH detection assay**. 2′,7′-dichlorodihydrofluorescein diacetate (DCFH-DA) and hydroxyphenyl fluorescein (HPF) were used to detect the intracellular ROS and •OH generation, respectively. For ROS generation study, 4T1 cells were cultured in a 96-well plate at 37 °C under hypoxic condition for 24 h. After different treatments, the cells were incubated with DCFH-DA (10 μM) in fresh medium for 20 min in the dark. The fluorescence intensities of DCFH-DA were measured by High Content System (PerkinElmer). For •OH generation study, 4T1 cells treated with PBS, PMNS and PMNSG containing HPF (10 μM) for 8 h, the fluorescence signals were measured by High Content System (PerkinElmer) in real-time.

**Investigation of tumor hypoxia amelioration**. 4T1 cells were cultured in 96-well plate. After different treatments, cells were fixed with 1% paraformaldehyde at 37 ºC for 15 min and permeabilized with PBS containing 0.2% Triton X-100 at 37 ºC for 5 min. After that, the blocking step was conducted by PBS with 0.05% Tween-20 buffer (containing 1% bovine serum albumin) at room temperature for 30 min. Subsequently, the cells were incubated with anti-α-Tubulin (1:1000, Cat. no. T9026, Sigma-Aldrich) and anti-HIF-1α (1:1000, Cat. no. 36169 T, Cell Signaling Technology) primary antibodies for 12 h, followed by labeling with secondary antibodies at room temperature for 1 h. Finally, the immunofluorescence intensities were analyzed by High Content System (PerkinElmer).

**Evaluation of lipid peroxidation**. For the lipid peroxidation evaluation, 4T1 cells were cultured in a 96-well plate at a density of $5 \times 10^3$ cells/well at 37 ºC with 5% $CO_2$ humidified atmosphere for 24 h. Subsequently, the culture medium was removed, and the cells were incubated with fresh medium containing PBS, GOx, PMNS and PMNSG, respectively, for 4 h. The 1064 nm laser was introduced to the corresponding well. After incubation for another 4 h, the cells were stained with Hoechst (1:1000) and BODIPY 581/591(1:1000) and detected by High Content System (PerkinElmer).

**Western blot**. The 4T1 Cells were lysed via vigorous sonication on ice bath and protein concentration was determined. Subsequently, immunoblotting of cell lysates was performed using antibodies to HIF-1α (1:1000, Cell Signaling Technology, catalogue no. 14179 S), VEGF (1:1000, Abcam, catalogue no. ab214424), caspase-3 (1:1000, Cell Signaling Technology, catalogue no.9662 S), GLUT1 (1:1000, Cell Signaling Technology, catalogue no. 12939 S) and β-actin (1:1000, Cell Signaling Technology, catalogue no. 8457 S) according to standard protocols. The proteins level was monitored by chemiluminescence imaging system (FluorChem E).

**ATP assay**. 4T1 cells were cultured in 6-well plate. After various treatments, the supernatants were collected, followed by treating with ATP assay kit (Beyotime Biotechnology, China). The ATP level was determined by enhanced chemiluminescence using microplate reader. The results presented are the average of three independent experiments.

**Tumor mouse model**. Female BALB/C nude mice aged 4-5 weeks were purchased from Guangdong Medicinal Laboratory Animal Center (Guangzhou, China) and All animal experiments were carried out in strict accordance with the regulations of the Animal Ethical and Welfare Committee of Shenzhen University (AEWC-SZU, maximal tumor size: <2000 mm³). All of the experimental mice were housed under standard conditions (temperature: ~22 ºC, humidity: 40–70%, 12 h dark-light cycles) with free access to sterile food and water. The 4T1 cells ($1 \times 10^6$ cell/site) in 100 μL PBS were subcutaneously injected into the right hindlimb of each mouse. When the tumor volume reached approximately 60 mm³ measured by a digital caliper, allowing for cancer therapy.

**PA imaging tracking**. NIR-II PA imaging were all performed on Vevo LAZR-Xsystem (VisualSonics Inc. New York, NY). For evaluating PA performance of PMNS and PMNSG in vitro, aqueous solution with gradient increasing concentration of the sample (0–60 ppm) were employed. The system setting parameters are as follows: frequency = 40 MHz, wavelength range = 1200–2000 nm, PA gain = 50 dB, gain = 13 dB depth/width = 10.00/14.08 mm and wavelength = 1261 nm.

For in vivo, 4T1 tumor-bearing mice ($n = 5$) were intravenously administrated with PMNS or PMNSG PBS solution (dose of 10 mg kg⁻¹), respectively. PA signals of the tumor sites were recorded before and after administration at prefixed time points (0, 1, 2, 4, 8, 12, 24, 48, 72 h). The system setting parameters are as follows: frequency = 40 MHz; wavelength range = 1200–2000 nm; PA gain = 49 dB; gain = 29 dB depth/width = 10.00/14.08 mm; wavelength = 1261 nm; 3D step size = 0.14 mm; scanning time: single-spectral PA imaging (about 1.5 min) and multispectral PA imaging (about 8 min); scanning area: ~450 mm²; pulse repetition rate: 20 Hz; safe energy: 20 mJ/cm².

In order to visualize the biodistribution and intensity proportion of PA signals in vivo. PMNS or PMNSG were intravenously injected into 4T1 tumor-bearing mice ($n = 3$). After 4 h post-injection, the tumor and major organs including heart, lung, liver, spleen and kidney were removed from euthanized mice and detected by VevoLAZR-X system to reconstruct the 3D model of PA signals of theses organs and tumors.

For PA monitoring the variation of OxyHb and DeoxyHb content in tumor tissue during therapy, the PA-Mode 3D (Multiwavelength) and PA-Mode 3D (Oxy-Hemo) of Vevo LAZR-X system were employed to record corresponding signals. The parameters of the system were listed as detailed in Supplementary Information (Supplementary Tables 1 and 2).

**In vivo biocompatibility examination**. For the long-term biocompatibility study, mice were divided into two groups ($n = 3$) randomly with different treatment: (1) Control group without any treatment, (2) PMNSG (dose of 10 mg kg⁻¹) intravenously injected into the mice. After 14 days of monitoring, the major organs (heart, spleen, kidney, liver, and lung) were collected after euthanasia of the mice. These organs were fixed in 4% paraformaldehyde and stained with H&E for histological analysis.

**In vivo cancer catalytic therapy**. The tumor-bearing mice were grouped randomly in seven ($n = 5$ per group): (a) Control; (b) GOx; (c) PMNS (d) NIR-II laser; (e) PMNS + NIR-II laser; (f) PMNSG; (g) PMNSG + NIR-II laser. GOx, PMNS and PMNSG in PBS solutions (injection dose = 100 μL, PMNS and GOx at dose of 10 and 0.2 mg kg⁻¹, respectively) were injected via the tail vein. For comparison, PBS (100 μL) was injected into mice as the control group. After 4 h of post-injection, under general anesthesia, tumor site of groups (d), (e) and (g) were exposed to NIR-II laser at the power density of 0.4 W cm⁻² for 10 min. At the same time, the thermal images were captured by an infrared thermal imager in real-time. After treatment, the tumor volumes were measured by a digital caliper or B-ultrasound module of Vevo-LAZR system, and the tumor volume was calculated by the equation: Volume = (Tumor length) × (Tumor width)²/2 (mm³) or by the Vevo-LAB built-in 3D reconstruction. Relative tumor volume was normalized to its initial size before the treatment. After two weeks, all mice were sacrificed, and tumors were collected and weighed.

**DCFH-DA staining of tumors**. The production of intratumor ROS were detected by DCFH-DA. The tumors with corresponding treatment dissected from euthanized mice were incubated with 10 μM DCFH-DA at 37 ºC in the dark for 30 min (Biyuntian, D6470, Solabio), then the tumors were washed with PBS and imaged within 20 min by in vivo fluorescence imaging system (PerkinElmer, USA). The specific fluorescence signals were collected at 525 nm.

**Histological studies and immunofluorescence**. After in vivo therapy and toxicity examination, all the mice were euthanized, the major organs and subcutaneous tumors were removed, washed with PBS and fixed in 4% fixative solution (Cat. no. P1110, Solarbio) and then embedded in paraffin. After that, the paraffin-embedded tumor sections were cut using a microtome (Leica RM2235, Germany) and mounted on slides. Finally, H&E-stained (G1005, Servicebio) tumor sections were stained using standard histological techniques. Anti-Ki-67 (catalogue no. GB111499, Servicebio), anti-Ly6G (catalogue no. GB112299, Servicebio), anti-HIF-1α (catalogue no.36169 T, Cell Signaling Technology), TUNEL (G1510, Servicebio), cleaved caspase-3 (catalogue no. 9661 S, Cell Signaling Technology) were employed for different sections staining, respectively. Finally, slices were photographed with a Virtual slide microscope (Olympus VS120, Japan). For immunofluorescence studies, the images of stained slices were obtained using a confocal laser scanning microscope (Zeiss LSM 710).

**Pearson correlation analysis among deoxyHb, oxyHb and PMNSG (or PMNS)**. Pearson product-moment correlation coefficient is a measure of the degree of correlation between two variables. It is a value between 1 and −1, where 1 means the variable is completely positively correlated, 0 means irrelevant, and −1 means completely negatively correlated. As illustrated in Supplementary Equation (11), where r is the Pearson correlation coefficient, x and y are the two variables among separated PA signals of OxyHb, DeoxyHb and PMNSG (or PMNS). The degree of correlation represented by the |r| value (Supplementary Table 3).

**Statistical analysis**. All data represent the mean ± SD. One-way ANOVA with Tukey's multiple comparisons was used for multiple comparisons when more than two groups were compared, and one-tailed or two-tailed Student's $t$-test was used for two-group comparisons. All statistical differences were calculated by using GraphPad Prism 8.0 (GraphPad Software, Inc., CA, USA).). In all types of statistical analysis values of $P < 0.05$ were considered significant.

**Reporting summary**. Further information on research design is available in the Nature Research Reporting Summary linked to this article.

## Data availability

The authors declare that the data supporting the findings of this study are available within the Article, Supplementary Information or Source data file. Source data are provided with this paper.

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

## Acknowledgements

This work is financially supported by National Key R&D Program of China (2018YFA0704000), National Natural Science Foundation of China (82071985), Basic Research Program of Shenzhen (JCYJ20180507182413022, JCYJ20170412111100742), Guangdong Province Natural Science Foundation of Major Basic Research and Cultivation Project (2018B030308003), Shenzhen Science and Technology Program (KQTD20190929172538530) and the Fok Ying-Tong Education Foundation for Young Teachers in the Higher Education Institutions of China (161032). We thank Instrumental Analysis Center of Shenzhen University (Lihu Campus).

## Author contributions

P.H. and S.L. conceived the concept and designed the project. S.L. and J.Z. performed the experiments and analyzed the results. N.B., M.L., D.-Y.Z., W.Y. and F.Z. assisted with the figure production and experiment design. P.H., J.L. and S.L. wrote and revised the original draft of the manuscript. All authors discussed the results and commented on the manuscript.

## Competing interests

The authors declare no competing interests.

**Additional information**

