## [Peer Review File · Nature Communications]

REVIEWER COMMENTS

Reviewer #1 (Remarks to the Author):

Review Comments:

The authors have represented a research article entitled “In vivo three-dimensional multispectral photoacoustic imaging of dual enzyme-driven cyclic cascade reaction for tumor catalytic therapy”. The authors represented a well-organized work with valuable outcomes. They conducted in vitro and in vivo tests to prove the effectiveness of this proposed nano-agent. The research study needs several corrections before acceptance.

1. Please try to explain the status of this experiment compared to the presently reported literature.
2. The stability of nanomaterials in serum is important for evaluating their potential application in vivo. So, the authors should detect the stability of their nanomaterials in serum (at least 50% FBS) to support their in vivo applications. In addition, the hydrodynamic size of the synthesized materials in PBS and FBS also should be provided.
3. What is the stability and distribution of the synthesized nanoparticles in body fluid/ media/ or in vivo systems?
4. Apart from subcutaneous xenograft tumors, the anti-cancer efficiency of synthesized nanomaterials should also be evaluated using the orthotopic tumor model in order to simulate the actual conditions of clinical cancer therapy.
5. The author adopted the manner of intravenous injection to evaluate the therapeutic effect, biodistribution, and imaging properties of nanomedicine in vivo. What are the target moieties authors choose to fabricate the nanomaterials? Please explain more elaborately.
6. In “Results” and in the Supplementary section “power X-ray” might be corrected as “powder X-ray”.

7. In figure 5 (e, f, and g), we find laser-induced the tumor grows faster. Bodyweight also increased, tumor weight also increased. Is there any relation between tumor growth and laser-induced? If so, then a combination of nanoparticles and laser may induce a mixed effect. Please explain.

8. In the PA experiment, several important information about the experimental setup is missing. For example, scanning time, scanning area, pulse repetition rate, and safe energy for in vivo experiments. Please incorporate that information.

Reviewer #2 (Remarks to the Author):

In this study, Huang and coworkers reported the use of 3D multispectral photoacoustic (PA) molecular imaging to monitor in vivo cascade catalytic therapy based on dual enzyme-driven cyclic reaction platform, which is consisted of 2D PdMo bimetallic nanosheet-based nanozyme conjugated with glucose oxidase (GOx). However, such nanomaterial of PMNS used for cancer treatment has recently been reported (J. Mater. Chem. B, DOI: 10.1039/d1tb01284c). Meanwhile, the idea of using PA imaging for monitoring molecular events during in vivo catalytic processes is not new, which has already been reported for monitoring the same molecular events of OxyHb and DeoxyHb in vivo (Photoacoustic molecular imaging-escorted adipose photodynamic–browning synergy for fighting obesity with virus-like complexes, Nat. Nanotechnol. 16, 455-465 (2021)). Therefore, although the work was well done and well written, the work may lack sufficient novelty to publish in Nature Communication, but some questions should be solved before publishing in other journals.

1. From Fig.1b, the absorption peak of PMNS was not obviously observed from UV-Vis-NIR spectra, compared with the reference reported (J. Mater. Chem. B, DOI: 10.1039/d1tb01284c). The authors need to explain it. And the error bars of Fig.1 c&f should be provided.

2. As we known, the 4T1 cells were cultured into DMEM which has high concentration of glucose of 25 mM. However, the GOx was modified on the surface of PMNS, which can easily react with glucose in DMEM under cell cultured process to produce large amounts of hydrogen peroxide to induce cell death, even though the PMNS-GOx particles were not entered into cells. How to solve this problem?

3. From Fig. S7, the TEM images of PMNS indicate that the dispersity of this nanomaterial is not good, even though the authors have provided the optical photos of PMNS dispersed in different physiological media. It's better to provide much more convincing data to prove the dispersability of PMNS.

4. As the authors described, the high tumor accumulation of PMNSG or PMNS may contribute to the passive targeting pathway of enhanced permeability and retention (EPR) effect. However, the most of PMNSG or PMNS were accumulated in liver, compared with tumor region (Fig. 3f). First, it can

affect the treatment effect. Secondly, most of PMNSG or PMNSM were accumulated in liver, which can impact on the normal metabolism of liver.

5. The optical photos of tumor-bearing mice before and after the treatment under different groups should be provided. The optical photos of tumors after the treatment also should be provided.

6. For cell viability test of the nanomaterials, the author tested more cell lines besides 4T1 cells (cf. Fig. S15), but they only chose 4T1 cells for treatment and PA imaging in the whole study. Why ?

7. The study (including title) focused on dual enzyme-driven cyclic cascade reaction for tumor catalytic therapy, the photothermal therapy of the nanomaterial may have stronger effect on cell apoptosis as the reported photothermal efficiency (temperature increase) of the PdMo-based nanomaterial was quite high.

8. Some relevant recent references on enzyme-driven cyclic cascade reaction should be cited, see for example, --- Tumor-selective catalytic nanomedicine by nanocatalyst delivery. *Nat. Commun.* 2017, 8, 357; Tumor Microenvironment-Activated Degradable Multifunctional Nanoreactor for Synergistic Cancer Therapy and Glucose SERS Feedback, *iScience*, 2020, 23, 101274; and Recent advances in glucose-oxidase-based nanocomposites for tumor therapy, *Small*, 2019, 15, 1903895.

Reviewer #3 (Remarks to the Author):

The authors developed a biodegradable cyclic cascade catalytic 65 system (denoted as PMNSG). Moreover, the system combined with GOx and PMNS enables noninvasively 3D PA tracking of dynamically molecular events associated with cascade catalytic process, thus providing feedback for precise tumor catalytic therapy. The aim is good, but the design is complex. Moreover, the catalytic process is dynamic, it's difficult to monitor the process using Imaging method. Although, the authors claim that the system is a catalytic system for cancer therapy, it's not a new approach for cancer theranostics. I do not recommend publishing the work.

1. For Figure 1a, the resolution is too low.

2. In vitro photothermal-enhanced cascade catalytic therapy, how did author choose the laser power?

3. The authors claimed that cascade catalytic process in living cells accompanied by the occurrence of multi-molecular events. But, for in vivo application, it's more complex. It's difficult to give a conclusion that living imaging can trace the process.

4. In this paper 10.1002/adma.201901778, many catalytic systems were reported. What's the novelty of the system in this work?

5. The molecular mechanism of PMNSG is not very clear. More experiments should be added to confirm the conclusion.

**Point-by-point response to Editor's and Reviewers' comments
for manuscript:**

“In vivo three-dimensional multispectral photoacoustic imaging of dual enzyme-driven cyclic cascade reaction for tumor catalytic therapy”

- Author responses are in **BLUE**

- Specific comments by the reviewers are **underlined**

Reviewer #1 (Remarks to the Author):

Review Comments:

The authors have represented a research article entitled “In vivo three-dimensional multispectral photoacoustic imaging of dual enzyme-driven cyclic cascade reaction for tumor catalytic therapy”. The authors represented a well-organized work with valuable outcomes. They conducted in vitro and in vivo tests to prove the effectiveness of this proposed nano-agent. The research study needs several corrections before acceptance.

Response: We truly thank the reviewer for the positive comments, and especially the acknowledgement of “a well-organized work with valuable outcomes” of our work.

1. Please try to explain the status of this experiment compared to the presently reported literature.

Response: Thanks for your comment. Firstly, compared to the presently reported studies¹, our work mainly focuses on the real-time non-invasive visualization of dynamic molecular events induced by cascade catalytic process via three dimensional (3D) photoacoustic (PA) molecular imaging in living system, thus enabling to deep our understanding of this biological process. Benefiting from the high optical contrast and high spatial resolution of PA imaging²⁻⁴, we obtained abundant tumor microenvironment information during the catalytic process. Importantly, the correlation between nanozyme states and the occurrence of molecular events in real-time was established for

the first time in this work via 3D PA imaging, which can be acted as feedback for precise tumor catalytic therapy.

In addition, most of current catalytic systems strongly depend on endogenous hydrogen peroxide (H_2O_2), while its concentration in tumor cells is rather low (less than $0.1 \mu\text{M}$)⁵, impeding the catalytic performance of nanozymes to result poor therapeutic efficacy. Therefore, we introduced glucose oxidase (GOx) enzyme, which can produce H_2O_2 through the aerobic catalysis of glucose into H_2O_2 and gluconic acid⁶. Of note, glucose plays a crucial role in cancer cell metabolism and proliferation (Warburg effect)⁷, and its depletion deprives the glycolysis pathway of its substrate, reducing the intracellular energy supply. Therefore, the GOx-mediated catalytic process can not only starve the tumors⁸, but the generated H_2O_2 is beneficial for tumor catalytic therapy. In consideration of the oxygen (O_2)-dependent catalytic process can be hindered by the hypoxic tumor microenvironment⁹, we prepared PdMo bimetallic nanosheets (PMNS) as the robust nanozyme to produce O_2 for GOx-triggered aerobic catalysis reaction. Compared to the majority of current Pd-based nanozymes, PMNS not only exhibited high CAT-like activity (under neutral or acidic conditions)^{10, 11} and POD-like activity (in an acidic medium)^{12, 13} to catalyze the decomposition of H_2O_2 into O_2 or highly cytotoxic hydroxyl radicals ($\cdot\text{OH}$), respectively, but also possessed second near-infrared (NIR-II) light induced photothermal effect to promote its catalytic activities, owing to the inherent localized surface plasmon resonance (SPR) effect¹⁴. More importantly, PMNS showed good biodegradability, which is suitable for biomedical applications. Therefore, the marriage of GOx and PMNS (PMNSG) is a complementary strategy to maximize catalytic therapeutic efficacy and minimize off-target toxicity.

On the basis of above analysis, the combination of 3D PA molecular imaging with efficient tumor catalytic therapy and glucose depletion using GOx conjugated PdMo bimetallic nanosheets holds great potential for advancing the feasibility of nanozymes for in vivo diagnosis and therapy.

2. The stability of nanomaterials in serum is important for evaluating their potential application in vivo. So, the authors should detect the stability of their nanomaterials in serum (at least 50% FBS) to support their in vivo

applications. In addition, the hydrodynamic size of the synthesized materials in PBS and FBS also should be provided.

Response: Thanks for your suggestion. We have characterized the stability of our nanomaterials during incubation with serum (60% FBS) for 15 days. As a result, our nanomaterials (PMNSG) showed excellent stability, which was confirmed by Dynamic Light Scattering (DLS) characterization, as shown in **Supplementary Fig. 14** in revised **Supplementary Information**. In addition, we also characterized the hydrodynamic size of PMNSG in PBS and FBS (**Supplementary Fig. 14**).

3. What is the stability and distribution of the synthesized nanoparticles in body fluid/ media/ or in vivo systems?

Response: Thanks for your comment. It is well known that the stability of nanomaterials is of great important for in vivo applications. Firstly, benefiting from the high optical contrast and high spatial resolution of 3D PA imaging, we mapped NIR-II PA signals to the tumors. As shown in **Fig. 4c and 4e**, after administration of PMNSG, the 3D NIR-II PA signals of PMNSG illuminated the entire tumor region, indicating the uniform intratumoral distribution of PMNSG as well as excellent stability in living system. Next, we also evaluated the biodistribution of PMNSG by ex vivo 3D PA imaging at 24 h post-injection. **As shown in Fig. 4f, 4g and Supplementary Fig. 30**, the 3D-rendered PA images revealed that the partial PMNSG accumulated in liver, followed by another organs, which were consistent with the results obtained via inductively coupled plasma massspectrometry (ICP-MS) (**Supplementary Fig. 31**). We hope that we were able to address this concern to the reviewer's satisfaction.

4. Apart from subcutaneous xenograft tumors, the anti-cancer efficiency of synthesized nanomaterials should also be evaluated using the orthotopic tumor model in order to simulate the actual conditions of clinical cancer therapy.

Response: Thanks so much for your constructive suggestion. The evaluation of anti-cancer efficiency of our system (PMNSG) on orthotopic tumor model is indeed beneficial to strengthen our manuscript. In this work, we mainly focus

on the combination of 3D PA imaging and cascade catalytic process, thus deepening our understanding the occurrence of dynamic molecular events in this biological process. We hope that this study will increase interest of researchers in the molecular events that accompanied by catalytic therapy in living system. In our future studies, we not only focus on the PA molecular imaging and molecular events, but also pay attention to the anti-cancer efficiency on different tumor model. We hope that we were able to address this concern to the reviewer's satisfaction.

5. The author adopted the manner of intravenous injection to evaluate the therapeutic effect, biodistribution, and imaging properties of nanomedicine in vivo. What are the target moieties authors choose to fabricate the nanomaterials? Please explain more elaborately.

Response: Thanks for your comment. Significantly, of the tumor targeting strategies, the enhanced permeability and retention (EPR) effect (passive targeting) of nanomedicines is a key mechanism for tumor targeting, and considered a gold standard for novel nanomedicines design. According to the previous reports^{15, 16}, the increased permeability of the blood vessels in tumors is characteristic of rapid and defective angiogenesis. Furthermore, the dysfunctional lymphatic drainage in tumors retains the accumulated nanomedicines. In vivo studies using nanomaterials of different mean size suggested that the threshold nanomaterial size for effective extravasation into tumors was less than 200 nm. In this context, we have prepared PdMo bimetallic nanosheets with a diameter of 56 nm (determined by TEM image), which is an appropriate size for passive targeting. Of note, As shown in Fig. 4c-4e, much obvious PA signals were clearly appeared in the tumor site (Fig. 5a, b), demonstrating the tumor uptake of PMNSG via the EPR effect. We hope that we were able to address this concern to the reviewer's satisfaction.

6. In "Results" and in the Supplementary section "power X-ray" might be corrected as "powder X-ray".

Response: Thanks for your kind reminding. We have corrected "power X-ray" to "powder X-ray" in revised manuscript and Supplementary Information, as marked with **BLUE** color.

7. In figure 5 (e, f, and g), we find laser-induced the tumor grows faster. Body weight also increased, tumor weight also increased. Is there any relation between tumor growth and laser-induced? If so, then a combination of nanoparticles and laser may induce a mixed effect. Please explain.

Response: We thank the reviewer for this comment. For in vivo tumor catalytic therapy, the tumor-bearing mice were grouped randomly in seven (n = 5 per group). As shown in revised Fig. 6e and Fig. 6f, the statistical significance was calculated with two-tailed Student's t test between laser and PBS groups. Obviously, there was no significant difference in tumor volume (p = 0.246) or tumor weight (p = 0.324) between laser and PBS groups. Therefore, there is no any relation between tumor growth and laser-induced. In addition, the body weights of mice after laser irradiation treatment were monitored for two weeks, and showed no obvious variations (**Fig. 6g**). Please see the **Fig. 6e-6g** in the manuscript.

8. In the PA experiment, several important information about the experimental setup is missing. For example, scanning time, scanning area, pulse repetition rate, and safe energy for in vivo experiments. Please incorporate that information.

Response: Thanks for your kind suggestion. For all PA imaging experiments, we have added scanning time, scanning area, pulse repetition rate, and safe energy in revised manuscript (**Methods: PA imaging tracking**) and Supplementary Information (**PA analysis of hemoglobin oxygen saturation**), as marked with **BLUE** color. The detailed information is also listed as follows:

Scanning time: single-spectral photoacoustic imaging (about 1.5 min);
multispectral photoacoustic imaging (about 8 min)

Scanning area: ~450 mm²

Pulse repetition rate: 20 Hz

Safe energy: 20 mJ/cm²

Reviewer #2 (Remarks to the Author):

Review Comments:

In this study, Huang and coworkers reported the use of 3D multispectral photoacoustic (PA) molecular imaging to monitor in vivo cascade catalytic therapy based on dual enzyme-driven cyclic reaction platform, which is consisted of 2D PdMo bimetallic nanosheet-based nanozyme conjugated with glucose oxidase (GOx). However, such nanomaterial of PMNS used for cancer treatment has recently been reported (J. Mater. Chem. B, DOI: 10.1039/d1tb01284c). Meanwhile, the idea of using PA imaging for monitoring molecular events during in vivo catalytic processes is not new, which has already been reported for monitoring the same molecular events of OxyHb and DeoxyHb in vivo (Photoacoustic molecular imaging-escorted adipose photodynamic–browning synergy for fighting obesity with virus-like complexes, Nat. Nanotechnol. 16, 455-465 (2021)). Therefore, although the work was well done and well written, the work may lack sufficient novelty to publish in Nature Communication, but some questions should be solved before publishing in other journals.

Response: We appreciate the reviewer's comments. Our responses are as below:

1. From Fig.1b, the absorption peak of PMNS was not obviously observed from UV-Vis-NIR spectra, compared with the reference reported (J. Mater. Chem. B, DOI: 10.1039/d1tb01284c). The authors need to explain it. And the error bars of Fig.1 c&f should be provided.

Response: Thanks for your comment. According to the previous report (J. Mater. Chem. B, DOI: 10.1039/d1tb01284c), we found that its preparation process of PdMo bimetallic was different with us. Concretely, Hou and co-workers prepared PdMo bimetallic by a high-pressure reaction using carbon monoxide (CO, 1 bar) as the reducing agent. However, our PdMo bimetallic nanosheets (PMNS) were prepared by a one-pot wet-chemical approach using ascorbic acid as reducing agent in a glass vial. The different preparation methods will result the various crystal structure and photophysical performance, especially for noble metal-based nanomaterials. Therefore, the absorption spectrum of PdMo bimetallic prepared by previous report (J. Mater. Chem. B, DOI: 10.1039/d1tb01284c) was different with our nanomaterials.

In addition, the extinction coefficient and peroxidase (POD)-like activity of PMNS have been characterized for three times, respectively, as shown in revised **Fig. 2c** and **Fig. 2f**. All changes were marked out with **BLUE** color.

2. As we known, the 4T1 cells were cultured into DMEM which has high concentration of glucose of 25 mM. However, the GOx was modified on the surface of PMNS, which can easily react with glucose in DMEM under cell cultured process to produce large amounts of hydrogen peroxide to induce cell death, even though the PMNS-GOx particles were not entered into cells. How to solve this problem?

Response: Thanks for your comment. As shown in **Supplementary Fig. 19**, the cell viability of 4T1 cells was evaluated after treated with various concentrations of GOx (equivalent with that of conjugating on PMNS) under normoxic condition. Obviously, the concentration of GOx needed to exceed 20 nM to cause obvious cytotoxicity. While, the conjugated GOx on the surface of PMNSG was determined to be about 0.1% of total mass via Bicinchoninic Acid Assay. In addition, the PMNS moiety of PMNSG showed excellent catalase (CAT)-like activity (**Fig. 2e**), which can efficiently catalyze the conversion of hydrogen peroxide (H_2O_2 , generated by GOx) into O_2 , thus further reducing toxicity of GOx even in DMEM with high concentration of glucose. Therefore, even if the nanoparticles do not enter the cell, they will not cause severe cell death. We hope that we were able to address this concern to the reviewer's satisfaction.

3. From Fig. S7, the TEM images of PMNS indicate that the dispersity of this nanomaterial is not good, even though the authors have provided the optical photos of PMNS dispersed in different physiological media. It's better to provide much more convincing data to prove the dispersibility of PMNS.

Response: Thanks for your kind suggestion. We have characterized the stability of our nanomaterials during incubation with various buffer solutions (H_2O , PBS, Saline, DMEM, FBS and Glucose) for 15 days. As a result, our nanomaterials (PMNSG) showed good stability. Please see **Supplementary Fig. 14** in the revised **Supplementary Information**. We hope that we were able to address this concern to the reviewer's satisfaction.

4. As the authors described, the high tumor accumulation of PMNSG or PMNS may contribute to the passive targeting pathway of enhanced permeability and retention (EPR) effect. However, the most of PMNSG or PMNS were accumulated in liver, compared with tumor region (Fig. 3f). First, it can affect the treatment effect. Secondly, most of PMNSG or PMNSM were accumulated in liver, which can impact on the normal metabolism of liver.

Response: Thank you for this comment. As shown in **Fig. 4f and 4g**, considerable amounts of nanomaterials (PMNSG) were accumulated in the tumor region. Furthermore, the blood circulation of PMNSG was also evaluated and exhibited a relatively long circulation time of 5.4 h (**Supplementary Fig. 32**). Fortunately, all of in vivo experimental results demonstrated that our cascade catalytic system could be efficiently worked in living system and have achieved our goals.

In addition, nanomaterials will inevitably be metabolized by the liver, and some of the nanomaterials will accumulate in the liver. How to reduce the accumulation of nanomaterials in the liver and its metabolism in the liver are also a research hotspot in recent years^{17, 18}. According to these studies, the severe hepatotoxicity or liver dysfunction was mainly resulted by the nondegradable nanomaterials. However, our nanomaterials (PMNSG) showed excellent degradability (**Supplementary Fig. 35**). In addition, there is no pathological abnormality or inflammation was observed in liver and even in other major organs (heart, spleen, lung and kidney) (**Supplementary Fig. 45**). Besides, even if the PMNSG has GOx on its surface, there is no obvious signal of Ly6G⁺ neutrophils could be observed in liver slice (**Supplementary Fig. 46**). These results indicated that the PMNSG possessed long-term biosafety, enabling to be used for in vivo applications. We hope that the responses are satisfactory to the reviewer and you would reconsider these experimental results in our manuscript.

5. The optical photos of tumor-bearing mice before and after the treatment under different groups should be provided. The optical photos of tumors after the treatment also should be provided.

Response: Thanks for your kind suggestion. We have provided the optical

photos of tumor-bearing mice before and after the treatment under different groups as well as tumors after the treatment. Please see **Supplementary Fig. 39** and **Supplementary Fig.41** in the revised Supplementary Information, as marked with **BLUE** color.

6. For cell viability test of the nanomaterials, the author tested more cell lines besides 4T1 cells (cf. Fig. S15), but they only chose 4T1 cells for treatment and PA imaging in the whole study. Why?

Response: Thank you for this comment. In our study, a series of cell lines were selected to test the cell viabilities, thus demonstrating that the PMNS possessed low cytotoxicity on normal cells (**Supplementary Fig. 16**). Among them, 4T1 cell line is a triple-negative breast cancer (TNBC) cell, a particularly aggressive subtype of the disease with worse outcomes and fewer treatment options. The development of effective therapeutics for TNBC may address an unmet medical need in breast cancer. Therefore, we have chosen the 4T1 cells and constructed 4T1-tumor bearing mice model for subsequent in vitro and in vivo investigation. However, we believe our system will exhibit similar performance on other types of tumor model because after incubation with PMNSG, the cancer cells showed higher ROS levels than those of normal cells (**Supplementary Fig. 22**), demonstrating the specificity of PMNSG toward cancer cells. In our future studies, we will conduct in multiple tumor models to further demonstrate the potential advantages of our researches.

7. The study (including title) focused on dual enzyme-driven cyclic cascade reaction for tumor catalytic therapy, the photothermal therapy of the nanomaterial may have stronger effect on cell apoptosis as the reported photothermal efficiency (temperature increase) of the PdMo-based nanomaterial was quite high.

Response: Thank you for this comment. As shown in **Fig. 3a and 3b**, even the 4T1 cells treated with high dosage of PMNS ($20 \mu\text{g mL}^{-1}$) plus laser irradiation (0.4 W cm^{-2}), the cell viabilities only showed a mild decrease. In addition, a mild tumor growth inhibition effect was observed in PMNS plus laser treatment group (**Fig. 6e**). These results indicated that photothermal therapy exhibited weak effect on cell apoptosis. In addition, the photothermal efficiency (temperature increase) of our nanomaterials (PMNSG: 0.4 W cm^{-2} ,

10 min, 48.2 °C) was equivalent with the **previous report** (*Nat. Commun.* **11**, 3712 (2020); Fig. 5e: 0.8 W cm⁻², 5 min, 50 °C)¹⁹, which also showed mild effect on its in vitro and in vivo experimental results (Ref 22: Fig. 6b). This is a strong evidence to further support our conclusions. We sincerely hope that the response is satisfactory to the reviewer and you would reconsider our experimental results.

8. Some relevant recent references on enzyme-driven cyclic cascade reaction should be cited, see for example, Tumor-selective catalytic nanomedicine by nanocatalyst delivery. *Nat. Commun.* 2017, 8, 357; Tumor Microenvironment-Activated Degradable Multifunctional Nanoreactor for Synergistic Cancer Therapy and Glucose SERS Feedback, *iScience*, 2020, 23, 101274; and Recent advances in glucose-oxidase-based nanocomposites for tumor therapy, *Small*, 2019, 15, 1903895.

Response: Thanks for your kind suggestion. We have cited these important references in revised manuscript (**Ref. 7, Ref. 8 and Ref. 9**), as marked with **BLUE** color. The cited references are also listed as follows:

7. Huo M, Wang L, Chen Y, Shi J. Tumor-selective catalytic nanomedicine by nanocatalyst delivery. *Nat. Commun.* **8**, 357 (2017).
8. Wang M, Wang D, Chen Q, Li C, Li Z, Lin J. Recent advances in glucose-oxidase-based nanocomposites for tumor therapy. *Small* **15**, 1903895 (2019).
9. Sun D, et al. Tumor microenvironment-activated degradable multifunctional nanoreactor for synergistic cancer therapy and glucose sers feedback. *iScience* **23**, 101274 (2020).

We would like to thank the reviewer for the constructive suggestions on how to further strengthen this manuscript. We have tried our best to address every comment in full and hope that our experiments and resulting data are satisfactory to the reviewer.

Reviewer #3 (Remarks to the Author):

Review Comments:

The authors developed a biodegradable cyclic cascade catalytic 65 system (denoted as PMNSG). Moreover, the system combined with GOx and PMNS

enables noninvasively 3D PA tracking of dynamically molecular events associated with cascade catalytic process, thus providing feedback for precise tumor catalytic therapy. The aim is good, but the design is complex. Moreover, the catalytic process is dynamic, it's difficult to monitor the process using Imaging method. Although, the authors claim that the system is a catalytic system for cancer therapy, it's not a new approach for cancer theranostics. I do not recommend publishing the work.

Response: We appreciate the reviewer's comments. Our responses are as below:

1. For Figure 1a, the resolution is too low.

Response: Thanks for your kind suggestions. We have provided the high-resolution images in the revised manuscript. Please see **Fig. 2a** in the revised manuscript.

2. In vitro photothermal-enhanced cascade catalytic therapy, how did author choose the laser power?

Response: Thank you for this comment. As shown in **Supplementary Fig. 6b**, upon 1064 nm laser irradiation with a power density of 0.4 W cm^{-2} for 5 min, the temperature of PMNS solution ($40 \mu\text{g mL}^{-1}$) could rise to about $50 \text{ }^\circ\text{C}$. Of note, according to our previous report²⁰, the GOx kept the maximum activity at about $50 \text{ }^\circ\text{C}$. In order to achieve the efficient monitoring of cascade catalytic process on cellular level and maximize the photothermal-enhanced cascade catalytic effect, the power density of 1064 nm laser is set as 0.4 W cm^{-2} . We sincerely hope that the response is satisfactory to the reviewer.

3. The authors claimed that cascade catalytic process in living cells accompanied by the occurrence of multi-molecular events. But, for in vivo application, it's more complex. It's difficult to give a conclusion that living imaging can trace the process.

Response: Thank you for this comment. Indeed, the living system is complex, that's why the cancer is so difficult to be treated completely. However, this is also the goal and motivation for scientific researches. To date, extensive efforts have been devoted to visualization of key multi-molecular events in

tumor evolution²¹⁻³², which can promote new cognition of tumor evolution and reveal the law of its occurrence and development, thus improving clinical diagnosis and treatment level, and promoting personalized diagnosis and treatment of tumor. For example, the immune response, molecular heterogeneity of breast cancer, drug-induced acute kidney injury and esophagitis carcinoma transformation have been efficiently visualized in vivo in real-time via NIR fluorescence imaging technology¹⁹⁻²², deepening our understanding of these pathological processes. Encouraged by these major breakthroughs, we conducted the non-invasive visualization of dynamic molecular events associated with tumor catalytic therapy in vivo via 3D PA molecular imaging. Benefiting from the high optical contrast and high spatial resolution of PA imaging, we obtained abundant tumor microenvironment information during the catalytic process (**Fig. 5, Supplementary Fig. 36 and Supplementary Fig. 37**). After careful analysis, we believe our experimental results are convincing to demonstrate that we can trace the catalytic process via 3D PA imaging. We sincerely hope that the response is satisfactory to the reviewer and you would reconsider our experimental results.

4. In this paper 10.1002/adma.201901778, many catalytic systems were reported. What's the novelty of the system in this work?

Response: Thanks for your comment. Firstly, compared to the presently reported studies (10.1002/adma.201901778)¹, our work mainly focuses on the real-time non-invasive visualization of dynamic molecular events induced by cascade catalytic process via three dimensional (3D) photoacoustic (PA) molecular imaging in living system, thus enabling to deep our understanding of this biological process. Benefiting from the high optical contrast and high spatial resolution of PA imaging²⁻⁴, we obtained abundant tumor microenvironment information during the catalytic process. Importantly, the correlation between nanozyme states and the occurrence of molecular events in real-time was established for the first time in this work via 3D PA imaging, which can be acted as feedback for precise tumor catalytic therapy.

In addition, most of current catalytic systems strongly depend on endogenous hydrogen peroxide (H₂O₂), while its concentration in tumor cells is rather low (less than 0.1 μM)⁵, impeding the catalytic performance of

nanozymes to result poor therapeutic efficacy. Therefore, we introduced glucose oxidase (GOx) enzyme, which can produce H_2O_2 through the aerobic catalysis of glucose into H_2O_2 and gluconic acid⁶. Of note, glucose plays a crucial role in cancer cell metabolism and proliferation (Warburg effect)⁷, and its depletion deprives the glycolysis pathway of its substrate, reducing the intracellular energy supply. Therefore, the GOx-mediated catalytic process can not only starve the tumors⁸, but the generated H_2O_2 is beneficial for tumor catalytic therapy. In consideration of the oxygen (O_2)-dependent catalytic process can be hindered by the hypoxic tumor microenvironment⁹, we prepared PdMo bimetallic nanosheets (PMNS) as the robust nanozyme to produce O_2 for GOx-triggered aerobic catalysis reaction. Compared to the majority of current Pd-based nanozymes, PMNS not only exhibited high CAT-like activity (under neutral or acidic conditions)^{10, 11} and POD-like activity (in an acidic medium)^{12, 13} to catalyze the decomposition of H_2O_2 into O_2 or highly cytotoxic hydroxyl radicals ($\bullet OH$), respectively, but also possessed second near-infrared (NIR-II) light induced photothermal effect to promote its catalytic activities, owing to the inherent localized surface plasmon resonance (SPR) effect¹⁴. More importantly, PMNS showed good biodegradability, which is suitable for biomedical applications. Therefore, the marriage of GOx and PMNS (PMNSG) is a complementary strategy to maximize catalytic therapeutic efficacy and minimize off-target toxicity.

On the basis of above analysis, the combination of 3D PA molecular imaging with efficient tumor catalytic therapy and glucose depletion using GOx conjugated PdMo bimetallic nanosheets holds great potential for advancing the feasibility of nanozymes for in vivo diagnosis and therapy.

5. The molecular mechanism of PMNSG is not very clear. More experiments should be added to confirm the conclusion.

Response: Thanks so much for your constructive suggestion. We have provided **lipid peroxidation (LPO) staining** and **Western blot** experiments to strengthen the molecular mechanism of our system (PMNSG) in revised manuscript (**Fig. 3h, Fig. 3j and Supplementary Fig. 28**). As shown in **Fig. 3g**, the PMNSG could efficiently induce the ROS generation, especially for $\bullet OH$, thus resulting the cell membrane damage due to the LPO³³. Obviously,

much stronger green fluorescence was observed in PMNSG-treated cells than the control group, and enhanced by 1064-nm laser irradiation (0.4 W cm^{-2}), suggesting that the endocytosed PMNSG triggered significant LPO in 4T1 cells during photothermal-enhanced cascade catalytic process (**Fig. 3h and Supplementary Fig. 28**). It is well established that hypoxia condition induces intracellular expression of hypoxia-inducible factor 1 α (HIF-1 α) protein³⁴. As revealed in **Fig. 3i**, the immunofluorescence staining (green fluorescence) images indicated that the generated O₂ can alleviate the hypoxic condition, resulting in the down-regulated expression of HIF-1 α protein in cancer cells. Given that HIF-1 α could induce the transactivation a variety of signal molecules, including vascular endothelial growth factor (VEGF) to increase angiogenesis and promote tumor growth⁹, we applied Western blot to investigate the expression of HIF-1 α and VEGF under different conditions (**Fig. 3j**). As a result, the PMNSG plus laser treatment group significantly induced the down-regulation of HIF-1 α and VEGF and exhibited a similar tendency as compared with immunofluorescence staining images (**Fig. 3i**), while negatively correlated with O₂ generation. Moreover, the efficient O₂ generation was beneficial for GOx-mediated aerobic catalysis toward glucose, thus causing severe glucose depletion to reduce the intracellular adenosine-5'-triphosphate (ATP) supply (**Fig. 3d**)³⁵. GLUT1, a glucose transporter protein, primarily regulates the cell uptake of glucose³⁶. The GOx-mediated aerobic catalysis could block the glycolysis in cells, down-regulating the GLUT1 expression to further decrease the cell uptake of glucose³⁷. Of note, the PMNSG showed higher negative regulation effect on GLUT1 expression, which could be further significantly down-regulated by laser irradiation (**Fig. 3j**). These results were strongly consistent with the ATP assay (**Fig. 3d**). Benefiting from the efficient cascade catalytic process, PMNSG can induce significant cell apoptosis with laser irradiation, as demonstrated by the obviously upregulated caspase-3 expression (**Fig. 3j**). Taken together, the summarized molecular mechanism of PMNSG plus laser-mediated catalytic therapy is shown in **Fig. 3k**. Our study identified that the combination of PMNS and GOx with 1064 nm laser irradiation significantly mediate the ATP level reduction, •OH level increase, HIF-1 α /VEGF/GLUT1 down-regulation and caspase-3 activation, thus efficiently inducing cell apoptosis.

Regarding the molecular mechanism that could be addressed with additional experiments, we hope that we were able to address this concern to the reviewer's satisfaction.

In summary, we thank the reviewer for the constructive suggestions which have further strengthened the manuscript. We have tried to address every comment to the best of our abilities and hope that the responses are satisfactory to the reviewer.

As an overall summary, we appreciate the very positive reception of this study by reviewer 1 and note that (s)he calls our work "a well-organized work with valuable outcomes". We hope that we were able to communicate convincing responses to the more general concerns of reviewer 2 and 3 that cannot be addressed via additional experiments. Regarding the comments that could be addressed with additional experiments, we hope that all three reviewers agree that we addressed them in full and in a clear fashion. The extensive new experiments, and the resulting data as illustrated in the **revised manuscript** and **Supplementary Information** have undoubtedly further strengthened the manuscript, and we hope that the reviewers now deem it suitable for publication in **Nature Communications**.

References

1. Yang B, Chen Y, Shi J. Nanocatalytic medicine. *Adv. Mater.* **31**, 1901778 (2019).
2. Guo B, *et al.* High-resolution 3D NIR-II photoacoustic imaging of cerebral and tumor vasculatures using conjugated polymer nanoparticles as contrast agent. *Adv. Mater.* **31**, 1808355 (2019).
3. Lin L, *et al.* High-speed three-dimensional photoacoustic computed tomography for preclinical research and clinical translation. *Nat. Commun.* **12**, 882 (2021).
4. Chen Y-S, Zhao Y, Yoon SJ, Gambhir SS, Emelianov S. Miniature gold nanorods for photoacoustic molecular imaging in the second near-infrared optical window. *Nat. Nanotechnol.* **14**, 465-472 (2019).
5. Giorgio M, Trinei M, Migliaccio E, Pelicci PG. Hydrogen peroxide: a metabolic by-product or a common mediator of ageing signals? *Nat Rev. Mol. Cell Bio.*

- 8**, 722-728 (2007).
6. Fu L-H, Qi C, Lin J, Huang P. Catalytic chemistry of glucose oxidase in cancer diagnosis and treatment. *Chem. Soc. Rev.* **47**, 6454-6472 (2018).
 7. Koppenol WH, Bounds PL, Dang CV. Otto Warburg's contributions to current concepts of cancer metabolism. *Nat. Rev. Cancer* **11**, 325-337 (2011).
 8. Fan W, *et al.* Glucose-responsive sequential generation of hydrogen peroxide and nitric oxide for synergistic cancer starving-like/gas therapy. *Angew. Chem. Int. Ed.* **56**, 1229-1233 (2017).
 9. Qiao Y, *et al.* Engineered algae: A novel oxygen-generating system for effective treatment of hypoxic cancer. *Sci. Adv.* **6**, eaba5996 (2020).
 10. Yang Y, *et al.* NIR-II driven plasmon-enhanced catalysis for a timely supply of oxygen to overcome hypoxia-induced radiotherapy tolerance. *Angew. Chem. Int. Ed.* **58**, 15069-15075 (2019).
 11. Li S, *et al.* Degradable holey palladium nanosheets with highly active 1d nanoholes for synergetic phototherapy of hypoxic tumors. *J. Am. Chem. Soc.* **142**, 5649-5656 (2020).
 12. Wang Q, Zhang L, Shang C, Zhang Z, Dong S. Triple-enzyme mimetic activity of nickel-palladium hollow nanoparticles and their application in colorimetric biosensing of glucose. *Chem. Commun.* **52**, 5410-5413 (2016).
 13. Xi Z, *et al.* Strain effect in palladium nanostructures as nanozymes. *Nano Lett.* **20**, 272-277 (2020).
 14. Huang X, *et al.* Freestanding palladium nanosheets with plasmonic and catalytic properties. *Nat. Nanotechnol.* **6**, 28-32 (2011).
 15. Peer D, Karp JM, Hong S, Farokhzad OC, Margalit R, Langer R. Nanocarriers as an emerging platform for cancer therapy. *Nat. Nanotechnol.* **2**, 751-760 (2007).
 16. Blanco E, Shen H, Ferrari M. Principles of nanoparticle design for overcoming biological barriers to drug delivery. *Nat. Biotechnol.* **33**, 941-951 (2015).
 17. Tsoi KM, *et al.* Mechanism of hard-nanomaterial clearance by the liver. *Nat. Mater.* **15**, 1212-1221 (2016).
 18. Cao M, *et al.* Molybdenum derived from nanomaterials incorporates into molybdenum enzymes and affects their activities in vivo. *Nat. Nanotechnol.* **16**, 708-716 (2021).
 19. Gong F, *et al.* Preparation of TiH_{1.924} nanodots by liquid-phase exfoliation for enhanced sonodynamic cancer therapy. *Nat. Commun.* **11**, 3712 (2020).
 20. He T, *et al.* Glucose oxidase-instructed traceable self-oxygenation/hyperthermia dually enhanced cancer starvation therapy.

- Theranostics* **10**, 1544-1554 (2020).
21. Fan Y, *et al.* Lifetime-engineered NIR-II nanoparticles unlock multiplexed in vivo imaging. *Nat. Nanotechnol.* **13**, 941-946 (2018).
 22. Huang J, Li J, Lyu Y, Miao Q, Pu K. Molecular optical imaging probes for early diagnosis of drug-induced acute kidney injury. *Nat. Mater.* **18**, 1133-1143 (2019).
 23. Zhong Y, *et al.* In vivo molecular imaging for immunotherapy using ultra-bright near-infrared-IIb rare-earth nanoparticles. *Nat. Biotechnol.* **37**, 1322-1331 (2019).
 24. Chen J, *et al.* Multiplexed endoscopic imaging of Barrett's neoplasia using targeted fluorescent heptapeptides in a phase 1 proof-of-concept study. *Gut*, **70**, 1010-1013 (2020).
 25. Stecker EC. Picturing inflammation in blood vessels. *Sci. Transl. Med.* **6**, 255ec166 (2014).
 26. Wang Z, *et al.* Two-way magnetic resonance tuning and enhanced subtraction imaging for non-invasive and quantitative biological imaging. *Nat. Nanotechnol.* **15**, 482-490 (2020).
 27. Wang C, *et al.* An electric-field-responsive paramagnetic contrast agent enhances the visualization of epileptic foci in mouse models of drug-resistant epilepsy. *Nat. Biomed. Eng.* **5**, 278-289 (2021).
 28. Warren W. An ultrasound bioprobe for biological imaging. *Science* **358**, 493 (2017).
 29. Farhadi A, Ho GH, Sawyer DP, Bourdeau RW, Shapiro MG. Ultrasound imaging of gene expression in mammalian cells. *Science* **365**, 1469 (2019).
 30. Weissleder R, Pittet MJ. Imaging in the era of molecular oncology. *Nature* **452**, 580-589 (2008).
 31. Provost J, *et al.* Simultaneous positron emission tomography and ultrafast ultrasound for hybrid molecular, anatomical and functional imaging. *Nat. Biomed. Eng.* **2**, 85-94 (2018).
 32. Momcilovic M, *et al.* In vivo imaging of mitochondrial membrane potential in non-small-cell lung cancer. *Nature* **575**, 380-384 (2019).
 33. He T, *et al.* Light-triggered transformable ferrous ion delivery system for photothermal primed chemodynamic therapy. *Angew. Chem. Int. Ed.* **60**, 6047-6054 (2021).
 34. Salceda S, Caro J. Hypoxia-inducible factor 1 α (HIF-1 α) protein is rapidly degraded by the ubiquitin-proteasome system under normoxic conditions: its stabilization by hypoxia depends on redox-induced changes. *J. Biol. Chem.*

- 272**, 22642-22647 (1997).
35. Vander Heiden MG, Cantley LC, Thompson CB. Understanding the Warburg Effect: the metabolic requirements of cell proliferation. *Science* **324**, 1029 (2009).
 36. Deng D, *et al.* Crystal structure of the human glucose transporter GLUT1. *Nature* **510**, 121-125 (2014).
 37. Shan L, *et al.* Organosilica-based hollow mesoporous bilirubin nanoparticles for antioxidation-activated self-protection and tumor-specific deoxygenation-driven synergistic therapy. *ACS Nano* **13**, 8903-8916 (2019).

REVIEWER COMMENTS

Reviewer #1 (Remarks to the Author):

The authors have made all the necessary corrections. The overall language of the manuscript is well written in this revised manuscript. The Data has been well represented and interpreted. The research conducted by the authors is significant for academic research. I would suggest possible publications in your esteemed journal.

Reviewer #2 (Remarks to the Author):

The manuscript has been carefully revised and almost satisfactorily addressed the comments and suggestions raised by the reviewer. But there are still some minor problems should be addressed before it can be published in Nature Communication.

1. The amount of GOx modified on the surface of PMNS should be provided. And as the authors said, the PMNS moiety of PMNSG showed excellent catalase (CAT)-like activity, which can efficiently catalyze the conversion of hydrogen peroxide (H_2O_2 , generated by GOx) into O_2 , thus further reducing toxicity of GOx even in DMEM with high concentration of glucose. However, how can the authors exclude the generation of oxygen radicals in the DMEM? because the generation of oxygen can convert to ROS under PMNS existed. The authors should explain or specify this.
2. The authors should provide the PDI values of PMNS dispersed in different physiological media to prove the dispersity of PMNS other than hydrodynamic size.
3. In the Figure 2d, we can find that the maximum temperature reaches to 70 oC from the photothermal curves of PMNS. How to ensure the activity of glucose oxidase? Authors should at least consider this or provide some data to verify this problem.

Reviewer #3 (Remarks to the Author):

In the revised manuscript, the authors have added additional experiments. However, the authors did not respond to the question of "What's the novelty of the work?". I agree with the reviewer #2 that I did not find the novelty of this work.

Moreover, except of the imaging results, no direct evidences was provided to check the correct of the results. Many direct experiments should be added.

It was reported that PA imaging was employed to monitor the events of xyHb and DeoxyHb in vivo (Nat. Nanotechnol. 16, 455-465 (2021)). This work is very similar with it. The idea of this work did not reach the requirement of Nature Communications.

I do not recommend to accepting the manuscript.

Point-by-point response to Reviewers' comments for manuscript:

“In vivo three-dimensional multispectral photoacoustic imaging of dual enzyme-driven cyclic cascade reaction for tumor catalytic therapy”

- Author responses are in **BLUE**

- Specific comments by the reviewers are **underlined**

Reviewer #1 (Remarks to the Author):

Review Comments:

The authors have made all the necessary corrections. The overall language of the manuscript is well written in this revised manuscript. The Data has been well represented and interpreted. The research conducted by the authors is significant for academic research. I would suggest possible publications in your esteemed journal.

Response: We truly thank the reviewer for the recognition of our work and revised manuscript.

Reviewer #2 (Remarks to the Author):

Review Comments:

The manuscript has been carefully revised and almost satisfactorily addressed the comments and suggestions raised by the reviewer. But there are still some minor problems should be addressed before it can be published in Nature Communication.

Response: We appreciate the reviewer's comments. Our responses are as below:

1. The amount of GOx modified on the surface of PMNS should be provided. And as the authors said, the PMNS moiety of PMNSG showed excellent catalase (CAT)-like activity, which can efficiently catalyze the conversion of hydrogen peroxide (H₂O₂, generated by GOx) into O₂, thus further reducing

toxicity of GOx even in DMEM with high concentration of glucose. However, how can the authors exclude the generation of oxygen radicals in the DMEM? because the generation of oxygen can convert to ROS under PMNS existed. The authors should explain or specify this.

Response: Thanks for your comment. We have characterized the amount of GOx modified on the surface of PMNS. As shown in **Extended Data Fig. 1**, the conjugated GOx on the surface of PMNSG was determined to be about 1.02% of total mass via Thermogravimetric Analysis (TGA), which was consistent with the result obtained from BCA Protein Assay.

In addition, Pd-based nanomaterials, a kind of excellent enzyme mimics, are exhibiting highly affinity for hydrogen peroxide (H_2O_2). Therefore, H_2O_2 is the specific and wide substrate for enzymatic reaction of Pd-based nanomaterials in biomedical applications¹⁻⁴. To date, only Xiong and coworkers reported that Pd-based nanomaterials could catalyze the oxygen (O_2) into singlet oxygen (1O_2)⁵. Significantly, this process mainly occurred on the Pd (100) facet not Pd (111) facet. However, in our study, the PMNS was mainly enclosed by Pd (111) facet (**Supplementary Fig. 1b**), which showed very weak affinity for O_2 . To further demonstrate the catalytic inertia of PMNS toward O_2 , 1,3-diphenylisobenzofuran (DPBF), a 1O_2 trapping reagent, was applied for the detection of 1O_2 in the solution of DMEM containing high concentration of glucose, H_2O_2 and PMNSG. As shown in **Extended Data Fig. 2**, negligible changes in the absorption intensity of DPBF can be seen, indicating that the O_2 could not be converted into ROS in this condition. We hope that we were able to address this concern to the reviewer's satisfaction.

Extended Data Fig. 1 TGA curves of PMNS and PMNSG powders under an air atmosphere with a heating rate of $10\text{ }^\circ\text{C min}^{-1}$.

Extended Data Fig. 2 Time-dependent absorption spectra of DPBF in the mixture of DMEM (high concentration of glucose), H₂O₂ (100 μM) and PMNSG (20 μg mL⁻¹).

2. The authors should provide the PDI values of PMNS dispersed in different physiological media to prove the dispersity of PMNS other than hydrodynamic size.

Response: Thanks for your comment. We have provided the PDI values of our nanomaterials in different physiological media, as shown in **Supplementary Table 4** in revised **Supplementary Information**. We hope that we were able to address this concern to the reviewer's satisfaction.

3. In the Figure 2d, we can find that the maximum temperature reaches to 70 °C from the photothermal curves of PMNS. How to ensure the activity of glucose oxidase? Authors should at least consider this or provide some data to verify this problem.

Response: Thanks for your comment. We have characterized the activity of GOx at different temperatures (30, 40, 50, 60 and 70 °C). As shown in **Extended Data Fig. 3**, the GOx exhibited maximal activity at 40 °C and 50 °C, followed by 30 °C and 60 °C, while it was indeed inactivated at 70 °C. However, to achieve 70 °C under laser irradiation, a dosage of 80 μg mL⁻¹ of the nanomaterials was required (**Fig. 2d**). For in vitro study, the maximal dosage of our nanomaterials used was 20 μg mL⁻¹ (**Fig. 3**), which is far below the 80 μg mL⁻¹. In addition, for in vivo study, the photothermal efficiency (temperature increase) of our nanomaterials was about 48.2 °C (**Fig. 6c**,

PMNSG: 0.4 W cm^{-2} , 10 min), which was close to the temperature of maximal activity of GOx ($50 \text{ }^{\circ}\text{C}$), while was far below the inactivated temperature ($70 \text{ }^{\circ}\text{C}$). Therefore, we were able to ensure the enzyme activity of GOx under the laser irradiation (0.4 W cm^{-2} , 10 min). We hope that we were able to address this concern to the reviewer's satisfaction.

Extended Data Fig. 3 Enzyme activity of GOx obtained by Hydrogen Peroxide Assay Kit (GOx: 10 ng mL^{-1} ; Glucose: 10 mM ; Time: 90 min) at different temperatures (30 , 40 , 50 , 60 and $70 \text{ }^{\circ}\text{C}$).

We would like to thank the reviewer for the constructive suggestions on how to further strengthen this manuscript. We have tried our best to address every comment in full and hope that our experiments and resulting data are satisfactory to the reviewer.

Reviewer #3 (Remarks to the Author):

Review Comments:

In the revised manuscript, the authors have added additional experiments. However, the authors did not respond to the question of “What’s the novelty of the work?”. I agree with the reviewer #2 that I did not find the novelty of this work. Moreover, except of the imaging results, no direct evidences were provided to check the correct of the results. Many direct experiments should be added. It was reported that PA imaging was employed to monitor the events of xyHb and DeoxyHb in vivo (Nat. Nanotechnol. 16, 455-465 (2021).

This work is very similar with it. The idea of this work did not reach the requirement of Nature Communications. I do not recommend to accepting the manuscript.

Response: We appreciate the reviewer's comments. Our responses are as below:

However, the authors did not respond to the question of "What's the novelty of the work?".

Response: Firstly, benefiting from the high optical contrast and high spatial resolution of photoacoustic (PA) imaging⁶⁻¹², especially for three dimensional (3D) PA molecular imaging¹³⁻¹⁴, we mainly focus on applying this imaging technology with great potential for the real-time non-invasive visualization of dynamic molecular events induced by cascade catalytic process in living system, thus enabling to deep our understanding of this biological process. However, 3D PA imaging has not yet been employed for monitoring the multi-molecular events during the cascade catalytic therapy. Next, although Pd-based nanomaterials have been reported previously for catalytic therapy^{1,2,4,5}, most of them show no near-infrared (NIR) absorption, thus do not have PA imaging capability. Finally, the PdMo bimetallic nanosheets (PMNS) show good biodegradability, which is the rare characteristic for noble-metal based nanozymes, possessing potent potential for biomedical applications.

It was reported that PA imaging was employed to monitor the events of xyHb and DeoxyHb in vivo (Nat. Nanotechnol. 16, 455-465 (2021)).

Response: The PA imaging was indeed employed to monitor the events of OxyHb and DeoxyHb in vivo according to the previous report (*Nat. Nanotechnol.* 16, 455-465 (2021))¹². However, the two dimensional (2D) PA imaging was the main imaging method applied in this work not 3D PA imaging. In comparison with 2D PA imaging, we can get more abundant biological information via 3D PA imaging. In addition, the above work focuses on the monitoring of the variation of OxyHb and DeoxyHb in adipose tissues during the photodynamic therapy, while our study mainly focuses on the monitoring of these molecular events in tumor tissues during the cascade catalytic process. We hope that our study will increase interest of researchers in the

molecular events that accompanied by catalytic therapy in living system.

As an overall summary, we appreciate the very positive reception of this study by reviewer 1 and reviewer 2. We hope that we were able to communicate convincing responses to the more general concerns of reviewer 2 and 3. Regarding the comments that could be addressed with additional experiments, we hope that reviewer 2 agrees that we addressed them in full and in a clear fashion. The extensive new experiments, and the resulting data have undoubtedly further strengthened the manuscript, and we hope that the reviewers now deem it suitable for publication in **Nature Communications**.

References

1. Yang Y, *et al.* NIR-II driven plasmon-enhanced catalysis for a timely supply of oxygen to overcome hypoxia-induced radiotherapy tolerance. *Angew. Chem. Int. Ed.* **58**, 15069-15075 (2019).
2. Li S, *et al.* Degradable holey palladium nanosheets with highly active 1d nanoholes for synergetic phototherapy of hypoxic tumors. *J. Am. Chem. Soc.* **142**, 5649-5656 (2020).
3. Wang Q, Zhang L, Shang C, Zhang Z, Dong S. Triple-enzyme mimetic activity of nickel–palladium hollow nanoparticles and their application in colorimetric biosensing of glucose. *Chem. Commun.* **52**, 5410-5413 (2016).
4. Xi Z, *et al.* Strain effect in palladium nanostructures as nanozymes. *Nano Lett.* **20**, 272-277 (2020).
5. Long R, *et al.* Surface facet of palladium nanocrystals: a key parameter to the activation of molecular oxygen for organic catalysis and cancer treatment. *J. Am. Chem. Soc.* **135**, 3200-3207 (2013).
6. Zhang Y, *et al.* Non-invasive multimodal functional imaging of the intestine with frozen micellar naphthalocyanines. *Nat. Nanotechnol.* **9**, 631-638 (2014).
7. Wang X, Pang Y, Ku G, Xie X, Stoica G, Wang LV. Noninvasive laser-induced photoacoustic tomography for structural and functional in vivo imaging of the brain. *Nat. Biotechnol.* **21**, 803-806 (2003).
8. Wang L, Maslov K, Wang LV. Single-cell label-free photoacoustic flowoxigraphy in vivo. *Proc. Natl Acad. Sci. USA* **110**, 5759 (2013).
9. Yao J, *et al.* High-speed label-free functional photoacoustic microscopy of mouse brain in action. *Nat. Methods* **12**, 407-410 (2015).
10. Hai P, *et al.* High-throughput, label-free, single-cell photoacoustic microscopy

- of intratumoral metabolic heterogeneity. *Nat. Biomed. Eng.* **3**, 381-391 (2019).
11. Wang LV, Hu S. Photoacoustic tomography: in vivo imaging from organelles to organs. *Science* **335**, 1458 (2012).
 12. Chen R, *et al.* Photoacoustic molecular imaging-escorted adipose photodynamic–browning synergy for fighting obesity with virus-like complexes. *Nat. Nanotechnol.* **16**, 455-465 (2021).
 13. Guo B, *et al.* High-resolution 3D NIR-II photoacoustic imaging of cerebral and tumor vasculatures using conjugated polymer nanoparticles as contrast agent. *Adv. Mater.* **31**, 1808355 (2019).
 14. Lin L, *et al.* High-speed three-dimensional photoacoustic computed tomography for preclinical research and clinical translation. *Nat. Commun.* **12**, 882 (2021).

REVIEWERS' COMMENTS

Reviewer #2 (Remarks to the Author):

The Authors responded to all my comments and I am happy to recommend this work for publication in Nature Communications.

**Point-by-point response to Reviewers' comments for
manuscript:**

“In vivo three-dimensional multispectral photoacoustic imaging of dual enzyme-driven cyclic cascade reaction for tumor catalytic therapy”

- Author responses are in **BLUE**
- Specific comments by the reviewers are **underlined**

Reviewer #2 (Remarks to the Author):

Review Comments:

The Authors responded to all my comments and I am happy to recommend this work for publication in Nature Communications.

Response: We truly thank the reviewer for the recognition of our work and revised manuscript.